# A Unified Framework with Environmental and Interaction Uncertainty for Robust Multi-Agent Reinforcement Learning

## Abstract

Multi-agent reinforcement learning (MARL) has achieved remarkable success across diverse domains, yet its robustness remains hindered by various inherent uncertainties arising from multi-agent systems. Although previous studies have explored robustness in MARL, most of them focus on a single type of uncertainty, without a unified framework to handle multiple sources simultaneously. As a result, their methods often fail to remain robust when exposed to diverse and interacting disturbances. To address this limitation, we propose a unified framework that explicitly models two complementary sources of uncertainty: environmental uncertainty, caused by stochastic dynamics, and interaction uncertainty, arising from the unpredictable behaviors of other agents. We capture these factors using hierarchical entropy-based uncertainty sets, which are then integrated into the robust Markov game formulation. This hierarchical design enables the framework to distinguish the distinct impacts of each uncertainty source while avoiding the excessive conservatism of treating them as a single unified set. On top of this formulation, we introduce the solution concept of an Aleatoric Robust Equilibrium (ARE), where each agent optimizes its policy against worst-case scenarios derived from the hierarchical sets. To compute the ARE, we first establish theoretical results on the existence and uniqueness of the robust value function and its convergence under value iteration. Based on these results, we develop tailored actor–critic algorithms for practical learning. Extensive experiments in both the multi-agent particle environment (MPE) and the multi-agent MuJoCo benchmark show that our approach achieves consistently superior robustness and performance across a wide range of uncertainty settings.

## 1 Introduction

MARL is essential for tackling complex decision-making tasks in dynamic and uncertain environments, with diverse applications such as autonomous driving (Dinneweth et al., 2022; Wu et al., 2022), robotics (Perrusquía et al., 2021; Zhong et al., 2025), strategic game-playing (Samvelyan et al., 2019), and resource allocation (Cui et al., 2020). However, real-world scenarios are inherently plagued by uncertainties stemming from stochastic dynamics, partial observability, and the non-stationarity of agent interactions (Kardeş et al., 2011; Li et al., 2014). Such uncertainties undermine the robustness and reliability of multi-agent systems, often resulting in degraded performance (Lockwood & Si, 2022). This motivates the development of robust MARL frameworks that can explicitly account for uncertainty and ensure reliable performance in complex environments.

Prior work on robust MARL has generally focused on a single source of uncertainty. On the one hand, methods involving environmental uncertainty focus on mitigating stochastic transitions or model noise to enhance policy stability under uncertain dynamics (He et al., 2023; Zhang et al., 2020b). On the other hand, methods involving interaction uncertainty focus on handling adversarial or stochastic perturbations in other agents' policies (Li et al., 2019; Muchen Sun et al., 2025). However, methods that consider only one source of uncertainty often fail in realistic multi-agent systems when both environmental and interactive perturbations occur simultaneously. This limitation underscores the need for a unified framework that jointly accounts for both types of uncertainty. Thus, we propose a unified framework that jointly models both sources of uncertainty. Fig. 1a depicts environmental uncertainty where obstacles emerge at random

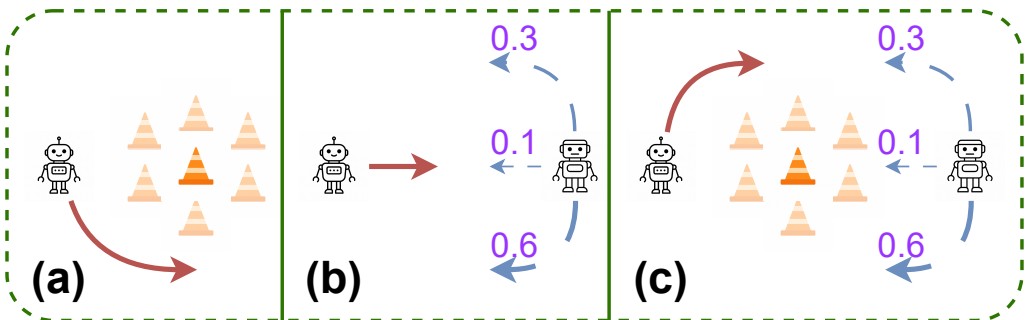

Figure 1: Illustration of two distinct sources of uncertainty in multi-agent systems. (a) Environmental uncertainty, arising from stochastic or uncertain obstacles in the environment; (b) Interaction uncertainty, induced by the stochastic and adaptive behaviors of other agents; (c) The proposed perspective considers both sources jointly, yielding a more realistic uncertainty representation for robust decision-making.

positions (transparent cones), requiring adaptive navigation. Fig. 1b highlights interaction uncertainty, where the right-side opponent chooses its motion probabilistically (e.g., preferring the middle path because it has a lower collision probability), thereby introducing behavioral unpredictability into the interaction. Fig. 1c combines both cases, showing that in practice agents must cope with uncertain environments and adaptive opponents. This joint perspective yields a more faithful representation of multi-agent uncertainty and enables more robust solutions.

In the robust MARL, uncertainty sets are commonly used to characterize worst-case realizations under which policies must remain safe. A straightforward approach is to combine multiple sources using a "max-of-both" construction. However, this formulation is problematic: the maximum environmental disturbance and the maximum interaction deviation rarely occur at the same time, yet the construction implicitly assumes their co-occurrence. As a result, the worst-case region is exaggerated, often leading to overly conservative policies (Xu et al., 2023; Huang et al., 2022). To address this, we introduce hierarchical entropy-based uncertainty sets, where interaction uncertainty is modeled at a higher level and environmental uncertainty at a lower level. This hierarchical design preserves the causal dependency between interaction and environmental uncertainty by conditioning environmental perturbations on the worst-case perturbed policy, rather than allowing independent maximization over both sources. Consequently, it maintains robustness while mitigating the excessive conservatism inherent in the conventional "max-of-both" formulation. We summarize the main contributions of this paper as follows:

- We propose a novel framework that incorporates *environmental* and *interaction* uncertainty components, enabling more targeted and effective mitigation in multi-agent reinforcement learning.

- We introduce a hierarchical structure for constructing uncertainty sets, which preserves consistency between different sources of uncertainty and leads to less conservative strategies.

- We define the ARE and establish theoretical results on the existence and uniqueness of the robust value function. Based on this formulation, we develop tailored actor–critic algorithms and validate the framework through extensive experiments across multiple benchmark environments.

## 2 Related Work

### 2.1 Aleatoric Uncertainty

Aleatoric uncertainty captures the inherent randomness in decision-making environments, such as stochastic state transitions, sensor noise, or unpredictable outcomes of actions. Unlike epistemic uncertainty, which can be reduced with more data, aleatoric uncertainty is irreducible and thus poses a fundamental challenge for

reliable reinforcement learning. Understanding and mitigating aleatoric uncertainty is particularly important in safety-critical and multi-agent systems, where such randomness can significantly undermine robustness. Recent research on robust RL under aleatoric uncertainty has mainly aimed to improve robustness and exploration efficiency (Machado et al., 2018; Moerland et al., 2017; Charpentier et al., 2022; Clements et al., 2019; Mavor-Parker et al., 2022). A variety of approaches have been proposed to handle this type of uncertainty in both exploration and exploitation. For example, Mai et al. (2022) proposes the IV-RL framework, which incorporates inverse-variance weighting to quantify aleatoric uncertainty, resulting in significant improvements in sample efficiency. Furthermore, Clements et al. (2019) develops a framework that disentangles aleatoric and epistemic uncertainties in Q-value predictions, enabling safer and more robust decision-making in risk-sensitive applications. In multi-agent systems, aleatoric uncertainty is more complex, as it stems from both the stochastic environment and the interdependent interactions among agents (Li et al., 2019; Tang et al., 2021; He et al., 2023; Amato & Oliehoek, 2013; Hao et al., 2023; Shi et al., 2024). This complexity motivates research on decomposing uncertainty into different sources, yet such work remains limited. Recent works such as Samvelyan et al. (2023) and Sharma et al. (2025) consider variations in both environment parameters and co-player behaviors during training, but they focus on curriculum generation and generalization rather than explicitly modeling uncertainty sets or decomposing the sources of aleatoric uncertainty. In contrast, our work studies the structure of aleatoric uncertainty in multi-agent systems and proposes a unified robust MARL framework that explicitly models and quantifies environmental and interaction uncertainties.

## 2.2 Robust Markov Games

Robust Markov game frameworks have been introduced to handle uncertainty and improve robustness (Aghassi & Bertsimas, 2006; Sharma & Gopal, 2007; Tan et al., 2019; Zhang et al., 2020a). Bukharin et al. (2024) proposes the ERNIE framework, which incorporates adversarial regularization to improve the robustness of policies in MARL. Xu et al. (2023) explores hierarchical latent variable models within the context of robust MARL to address over-conservatism in traditional robust methods. McMahan et al. (2024) investigates the computational complexity in robust Markov games with $s$-rectangular uncertainty and establishes equivalences between robust Nash equilibria and regularized Markov games. Ma et al. (2023) examines robustly correlated equilibria in Markov games with model uncertainty and introduces a robust decentralized V-learning algorithm that achieves polynomial sample complexity. In our work, we introduce a robust Markov game formulation that explicitly accounts for both sources of aleatoric uncertainty through hierarchical entropy-based sets.

# 3 Methodology

In this section, we present our robust MARL framework. We first **formulate the problem as a robust Markov game**, where uncertainties from both the environment and agent interactions are explicitly modeled. To capture these uncertainties effectively, we introduce **hierarchical entropy-based uncertainty sets**, which naturally lead to the solution concept of the ARE, which is defined as a state where each agent maximizes its return under worst-case scenarios given by the hierarchy. Building on this formulation, we then **develop tailored policy gradient and actor–critic algorithms** that can effectively compute the ARE in practice, together with the implementation details on constructing entropy-based uncertainty sets and applying them during learning.

## 3.1 Problem Formulation

### 3.1.1 Markov Games

A standard Markov game Littman (1994) is defined as a tuple:

$$\mathcal{G} = \langle \mathcal{S}, N, \{\mathcal{A}_i\}_{i=1}^N, \{R_i\}_{i=1}^N, P, \gamma \rangle. \tag{1}$$

where $\mathcal{S}$ is the state space, and $N$ is the number of agents. For each agent $i$, $\mathcal{A}_i$ is the action space, and the joint action space is $\mathcal{A} = \mathcal{A}_1 \times \cdots \times \mathcal{A}_N$. $R_i : \mathcal{S} \times \mathcal{A} \to \mathbb{R}$ represents the reward function for agent $i$.

$P : \mathcal{S} \times \mathcal{A} \to \Delta(\mathcal{S})$ is the state transition function, where $\Delta(\mathcal{S})$ denotes the probability distribution over the state space. And $\gamma \in (0, 1]$ is the discounting factor. The notations throughout this paper are summarized in Appendix A.

At each timestep, each agent $i$ selects an action $a_i \in \mathcal{A}_i$ according to its policy $\pi_i \in \Pi_i$, forming a joint action $a = (a_1, \ldots, a_N) \in \mathcal{A}$ and a joint policy $\pi = (\pi_1, \ldots, \pi_N)$, where $\pi : \mathcal{S} \to \mathcal{A}$. The joint action leads to the next state $s'$ according to the transition function $P(s' \mid s, a)$. Each agent $i$ receives an individual reward $r_i = R_i(s, a)$ and learns to maximize its value function:

$$V_i(s) = \mathbb{E}\left[ \sum_{t=0}^{\infty} \gamma^t R_i(s^t, a^t) \mid s_0 = s, a^t \sim \pi, P \right] = \mathbb{E}\left[ R_i(s, a) + \gamma V_i(s') \right], \tag{2}$$

where $s_0$ is the initial state.

### 3.1.2 Robust Markov Games

Inspired by the works (Satia & Lave Jr, 1973; Aghassi & Bertsimas, 2006; Nilim & Ghaoui, 2003), we extend standard Markov games to robust Markov games, which consider two different types of uncertainty resources. The entropy-based robust Markov Games are defined as a tuple:

$$\mathcal{G}_R = \langle \mathcal{S}, N, \{\mathcal{A}_i\}_{i=1}^N, \{\mathcal{R}_i\}_{i=1}^N, \tilde{\mathcal{P}}_s, \{\Delta_{\pi_i}\}_{i=1}^N, \gamma \rangle, \tag{3}$$

where $\mathcal{S}$, $N$, $\mathcal{R}_i$, and $\gamma$ represent the state space, the number of agents, the reward function for agent $i$, and the discount factor, respectively. $\mathcal{A}_i$ denotes the action space for the agent $i$, where actions are sampled from the uncertainty-perturbed policy $\bar{\pi}_i$. The policy $\bar{\pi}_i$ is defined as $\bar{\pi}_i = \pi_i + \delta_i$, where $\delta_i \in \Delta_{\pi_i}$ represents the perturbation of the policy of the set of interaction uncertainties $\Delta_\pi = \{\Delta_{\pi_i}\}_{i=1}^N$. In addition, $\tilde{\mathcal{P}}_s$ represents the environmental uncertainty sets for the transition probabilities. We distinguish between two types of transition kernels: $P(s'|s, a)$ is nominal transition model, $\bar{P}(s'|s, a)$ is perturbed transition kernel selected by the adversary.

In robust Markov games, adversarial learning has been widely explored to enhance robustness, following prior works (Subramanian et al., 2023; Guo et al., 2022; Aghassi & Bertsimas, 2006). A common approach is to introduce **nature players** that actively search for worst-case realizations within the predefined uncertainty sets. By identifying these adversarial scenarios that minimize cumulative rewards, the normal agents are then trained to maximize their strategies against them, leading to a principled min–max optimization formulation.

**Definition 3.1.** The goal for robust Markov Games is to learn a value function $\bar{V}_i^*(s)$ by updating the Bellman equation recursively. The optimization is taken over the policy of agent $i$, while the other agents are assumed to follow their optimal strategies. Specifically, for agents $j \neq i$, where $\bar{\pi}_j^* = \pi_j^* + \delta_j^*$ denotes the optimal perturbed policy, $\pi_j^*$ denotes their optimal policies, $\delta_j^*$ represents the corresponding interaction disturbances, and $a_j$ is sampled following $a_j \sim \bar{\pi}_j^*$.

$$\begin{aligned} \bar{V}_i^*(s) &= \max_{\pi_i} \min_{\substack{\delta_i \in \Delta_{\pi_i} \\ \bar{P} \in \tilde{\mathcal{P}}_s(\pi)}} \mathbb{E}\left[ \sum_{t=0}^{\infty} \gamma^t R_i(s^t, a^t) \mid s_0 = s, \pi, \delta_i, \bar{P} \right] \\ &= \max_{\pi_i} \min_{\substack{\delta_i \in \Delta_{\pi_i} \\ \bar{P} \in \tilde{\mathcal{P}}_s(\pi)}} \sum_{a_i \in \mathcal{A}_i} \pi_i(a_i \mid s) \prod_{j \neq i} \bar{\pi}_j^*(a_j \mid s, \eta_j^*) \left( R_i(s, a) + \gamma \sum_{s' \in \mathcal{S}} \bar{P}(s' \mid s, a) \bar{V}_i^*(s') \right). \end{aligned} \tag{4}$$

**Theorem 3.2** (Existence and Uniqueness of the Robust Value Function). *Assume that the state space $\mathcal{S}$ and action space $\mathcal{A}$ are bounded, the uncertainty sets $\Delta_{\pi_i}$ and $\tilde{\mathcal{P}}_s$ are compact and non-empty. And the robust value function $\bar{V}_i(s)$ satisfies the Bellman operator $T$:*

$$T[\bar{V}_i](s) = \max_{\pi_i} \min_{\substack{\delta_i \in \Delta_{\pi_i} \\ \bar{P} \in \tilde{\mathcal{P}}_s(\pi)}} \sum_{a_i \in \mathcal{A}_i} \pi_i(a_i \mid s) \prod_{j \neq i} \bar{\pi}_j^*(a_j \mid s, \eta_j^*) \left( R_i(s, a) + \gamma \sum_{s' \in \mathcal{S}} \bar{P}(s' \mid s, a) \bar{V}_i(s') \right). \tag{5}$$

*Then, there exists a unique value function $\bar{V}_i^*$ such that*

$$T[\bar{V}_i^*](s) = \bar{V}_i^*(s), \quad \forall s \in \mathcal{S}, \ i \in \mathcal{N},$$

*i.e., $\bar{V}_i^*$ is the unique fixed point of the Bellman operator $T$.*

The proof of these properties is provided in Appendix B.1.

### 3.2 Aleatoric Uncertainty Decomposition

As introduced in the problem formulation, we explicitly consider two types of aleatoric uncertainty in multi-agent systems: environmental uncertainty arising from stochastic environment dynamics, and interaction uncertainty caused by the unpredictable behaviors of other agents. One of the objectives of this work is to explicitly characterize and quantify these two sources of uncertainty. To this end, we analyze the structure of uncertainty using entropy decomposition. Specifically, by applying the chain rule of entropy, the conditional entropy of the next state can be written as Aleatoric uncertainty in multi-agent systems arises from two fundamental sources:

$$H(S' \mid s) = \underbrace{\mathbb{E}_{a \sim \pi(a|s)}[H(S' \mid s, a)]}_{\substack{\text{Environmental} \\ \text{Uncertainty}}} + \underbrace{H(a \mid s)}_{\substack{\text{Interaction} \\ \text{Uncertainty}}}, \tag{6}$$

where the $H(a \mid s) = -\sum_{a \in \mathcal{A}} \pi(a \mid s) \log \pi(a \mid s)$, and $S'$ denotes the random variable of the next state. Eq. 6 decomposes the total conditional entropy of the next state into two complementary sources which together provide a complete characterization of aleatoric uncertainty in multi-agent systems. The first term $\mathbb{E}_{a \sim \pi(a|s)}[H(S' \mid s, a)]$, corresponds to *environmental uncertainty*, which reflects the randomness of state transitions when the current state and action are fixed. The second term, $H(a \mid s)$, corresponds to *interaction uncertainty*, which captures the stochasticity of agents' action selections given the same state.

### 3.3 Hierarchical Entropy-based Uncertainty Sets

In the max-min optimization framework, selecting perturbations from the uncertainty sets is critical for ensuring robustness. In this section, we describe the construction of hierarchical entropy-based uncertainty sets. Building on the entropy formulation in Eq. 6, we observe a natural hierarchy: the randomness in action selection directly affects the distribution of future states. This motivates us to analyze the uncertainty in two stages.

At the higher level, we define the **interaction uncertainty set** that models perturbations in agents' strategies. Specifically, given the nominal policy $\pi_i$, the perturbed policy $\bar{\pi}_i = \pi_i + \delta_i$ belongs to

$$\Delta_{\pi_i} = \left\{ \delta_i \mid H(a \mid s) - H_{\bar{\pi}_i}(\bar{a} \mid s) \leq \rho_{\text{int}}, \forall a \sim \pi, \forall \bar{a} \sim \bar{\pi} \right\}, \tag{7}$$

where $H_{\bar{\pi}_i}(\bar{a} \mid s)$ denotes the entropy of the perturbed policy, and $\rho_{\text{int}}$ bounds the admissible deviation in action entropy. Conditioned on the worst-case perturbed policy $\bar{\pi}$ drawn from $\Delta_{\pi_i}$,

we define the lower-level **environmental uncertainty set** as

$$\tilde{\mathcal{P}}_s(\bar{\pi}) = \left\{ \bar{P}^H(\cdot \mid s, a, \delta) \mid \left| H(S' \mid s, a) - H_{\bar{P}^H}(S' \mid s, \bar{a}) \right| \leq \rho_{\text{env}}, \forall a \sim \pi, \forall \bar{a} \sim \bar{\pi} \right\} \tag{8}$$

where $\rho_{\text{env}}$ controls the permissible variation in transition entropy. The term $H(S' \mid s, a)$ denotes the entropy under the nominal transition dynamics $P(\cdot \mid s, a)$ without disturbances. Here $\bar{P}^H$ denotes the adversarial transition kernel selected by the nature player under the hierarchical uncertainty structure. Although the hierarchical disturbed transition probability $\bar{P}^H(\cdot \mid s, a, \delta)$ is defined with respect to the conditional disturbance $\delta$, its influence is reflected implicitly through the perturbed action distribution $\bar{\pi}$ rather than by explicitly augmenting the dynamics model with $\delta$. Together, these two nested sets form the overall **hierarchical uncertainty set**:

$$\mathcal{U}_H = \left\{ (\bar{P}^H, \delta) \mid \delta \in \Delta_\pi, \ \bar{P}^H(s' \mid s, a, \delta) \in \tilde{\mathcal{P}}_s(\bar{\pi}), \forall s, s' \in \mathcal{S}, \forall a \sim \pi \right\}, \tag{9}$$

which captures the causal dependence between interaction and environmental uncertainties. For comparison, we also define the general uncertainty set $\mathcal{U}_G$, which allows environmental and interaction disturbances to be selected independently:

$$\mathcal{U}_G = \left\{ (\bar{P}, \delta) \mid \delta \in \Delta_\pi, \ \bar{P}(s' \mid s, a) \in \tilde{\mathcal{P}}_s(\pi), \forall s, s' \in \mathcal{S}, \forall a \sim \pi \right\}. \tag{10}$$

In contrast to $\mathcal{U}_H$, the set $\mathcal{U}_G$ does not impose any dependency between the interaction perturbation $\delta$ and the environmental transition kernel $\bar{P}$. As a result, the adversary can select disturbances independently from the product of the two sets.

**Lemma 3.3** (Subset Property). *The hierarchical uncertainty set $\mathcal{U}_H$ is a subset of the general product set $\mathcal{U}_G$ where $\mathcal{U}_G$ allows nature to select worst-case interaction and environmental perturbations independently.*

$$\mathcal{U}_H \subseteq \mathcal{U}_G. \tag{11}$$

The proof can be found in Appendix B.2.

**Theorem 3.4** (Performance Bound Improvement). *For any policy $\pi$ and state $s$, the worst-case value under $\mathcal{U}_H$ is no worse than that under $\mathcal{U}_G$:*

$$\inf_{(\bar{P}^H, \delta) \in \mathcal{U}_H} V_{\bar{P}^H, \delta}(s) \ \geq \ \inf_{(\bar{P}, \delta) \in \mathcal{U}_G} V_{\bar{P}, \delta}(s). \tag{12}$$

The proof is listed in Appendix B.3.

Together, Lemma 3.3 and Theorem 3.4 show that the hierarchical sets exclude unrealistic joint disturbances and provide stronger robustness guarantees.

With the hierarchical uncertainty sets in place, we now need to specify the corresponding solution concept. To this end, we define the **ARE**, which characterizes the equilibrium outcome of agents optimizing against the worst-case disturbances derived from these sets.

**Definition 3.5.** Consider the optimal policy of the nature player as $\eta^* = (\eta_1^*, \eta_2^*, \ldots, \eta_N^*)$, where $\eta$ represents the coupling policies of $\bar{P}^H$ and $\delta$. A joint policy $\pi^* = (\pi_1^*, \pi_2^*, \ldots, \pi_N^*)$ and $\eta^*$ together form the ARE. If for any $s \in \mathcal{S}$ and all $i \in \mathcal{N}$, there exists a vector of value functions $V_* = (\bar{V}_1^*, \ldots, \bar{V}_N^*)$, with each $\bar{V}_i^* : \mathcal{S} \to \mathbb{R}$, satisfying:

$$(\pi_i^*(\cdot \mid s), \eta_i^*(\cdot \mid s)) \in \arg\max_{\pi_i(\cdot \mid s)} \min_{\eta_i(\cdot \mid s)} \sum_{a_i \in \mathcal{A}_i} \pi_i(a_i \mid s) \prod_{j \neq i} \bar{\pi}_j^*(a_j \mid s, \eta_j^*) \left( R_i(s, a) + \gamma \sum_{s' \in \mathcal{S}} \bar{P}^H(s' \mid s, a, \delta_i) \bar{V}_i^*(s') \right),$$
$$\tag{13}$$

where $\bar{\pi}_j^* = \pi_j^* + \delta_j^*$ denotes the optimal perturbed policy of agent $j$, and $a_j$ is sampled following $a_j \sim \bar{\pi}_j^*$.

**Theorem 3.6** (Existence of ARE). *Assume that $\mathcal{S}$ and $\mathcal{A}$ are finite sets, $\Delta_{\pi_i}$ and $\tilde{\mathcal{P}}_s(\bar{\pi})$ are compact and convex, and the reward function $R_i(s, a)$ and transition probability $\bar{P}^H(s' \mid s, a, \delta)$ are bounded. Then the ARE exists.*

The proof can be found in Appendix B.4. Unlike a standard Nash equilibrium, the ARE is a robust Markov perfect Nash equilibrium, where the nature player acts as a state-wise adversary constrained by hierarchical entropy-based uncertainty sets, rather than as a strategic long-horizon opponent.

### 3.4 Actor-Critic Update for ARE

To achieve the ARE, it is essential to adopt effective update methods capable of finding this equilibrium. In this part, we introduce the actor-critic approach as a means of achieving ARE. Specifically, we first define a **parametric goal function** that captures the optimization of each agent under the hierarchical uncertainty sets. We then introduce **parameterized policy update rules** for both the agent and its corresponding nature players, enabling gradient-based learning in the min–max setting. Finally, we derive and implement the **actor–critic update rule**, which jointly learns the actor and the robust critic through temporal-difference learning.

### 3.4.1 Parametric Goal Function

Each agent $i \in \mathcal{N}$ has a policy $\pi_i$ parameterized by $\pi_{\theta_i}$. The corresponding adversarial strategy, which governs the perturbation of the policy, is denoted as $\delta_i$ and parameterized by $w_i$, i.e., $\delta_{w_i}$. Additionally, the transition dynamics for agent $i$ are governed by $\bar{P}_i$, parameterized by $\xi_i$, denoted as $\bar{P}_{\xi_i}$.

The overall parametric joint strategy, including both agents and nature players, is represented as:

$$\tilde{\pi}_\theta = (\pi_{\theta_1}, \delta_{w_1}, \bar{P}^H_{\xi_1}, \pi_{\theta_2}, \delta_{w_2}, \bar{P}^H_{\xi_2}, \ldots, \pi_{\theta_N}, \delta_{w_N}, \bar{P}^H_{\xi_N}). \tag{14}$$

We define the objective of each agent $i$ under the joint strategy $\tilde{\pi}_\theta$ as the expected cumulative reward: $J_i(\theta) := \bar{V}^i_{\tilde{\pi}_\theta}(s_0)$. And the parametric robust value function satisfies the following recursive relationship:

$$\bar{V}^i_{\tilde{\pi}_\theta}(s) = \sum_{a_i \in \mathcal{A}_i} \pi_{\theta_i}(a_i \mid s) \prod_{j \neq i} \bar{\pi}_{\theta_j}(a_j \mid s, \delta_{w_j}) \left( R_i(s, a) + \gamma \sum_{s' \in \mathcal{S}} \bar{P}^H_{\xi_j}(s' \mid s, a, \delta) \bar{V}^i_{\tilde{\pi}_\theta}(s') \right), \tag{15}$$

where $\bar{\pi}_{\theta_j} = \pi_{\theta_j} + \delta_{w_j}$. Similarly, we define the parametric Q-value function with parameters $\psi_i$, which satisfies the fixed-point equation:

$$Q_{\psi_i}(s, a) = R_i(s, a) + \gamma \sum_{s' \in \mathcal{S}} \bar{P}^H_{\xi_i}(s' \mid s, a, \delta) \sum_{a'_i \in \mathcal{A}_i} \pi_{\theta_i}(a'_i \mid s') \prod_{j \neq i} \bar{\pi}_{\theta_j}(a'_j \mid s', \delta_{w_j}) Q_{\psi_i}(s', a'), \tag{16}$$

where $a'$ denotes action of next step.

**Theorem 3.7** (Convergence of Hierarchical Robust Q-value). *The hierarchical robust Bellman operator $\mathcal{T}^H$ defined over the uncertainty set $\mathcal{U}_H$ is a $\gamma$-contraction mapping in the $L_\infty$ norm. That is, for any two Q-functions $Q^k_i, Q^{k+1}_i$, it holds that:*

$$\|\mathcal{T}^H Q^k_i - \mathcal{T}^H Q^{k+1}_i\|_\infty \leq \gamma \|Q^k_i - Q^{k+1}_i\|_\infty$$

*Consequently, the sequence $Q^{k+1}_i = \mathcal{T}^H Q^k_i$ converges to a unique fixed point $Q^*_i$, representing the optimal robust Q-value.*

The proof is provided in Appendix B.6

### 3.4.2 Parametric Policy Update Rules

Given the objective defined above, we now derive the corresponding policy gradient updates that enable each agent to optimize its policy. For each agent $i$, the policy gradient of the objective $J_i(\theta)$ with respect to the parameter $\theta_i$ is given by:

$$\nabla_{\theta_i} J_i(\theta) = \mathbb{E}_{s,a \sim \zeta_{\tilde{\pi}_\theta}} \left[ \frac{\pi_{\theta_i}(a_i \mid s)}{\pi_{\theta_i}(a_i \mid s) + \delta_{w_i}(a_i \mid s)} \nabla_{\theta_i} \log \pi_{\theta_i}(a_i \mid s) \cdot Q_{\psi_i}(s, a) \right], \tag{17}$$

where $\zeta_{\tilde{\pi}_\theta}$ is the discounted state visitation measure Pr under the joint strategy $\tilde{\pi}_\theta$, defined as: $\zeta^{s_0}_{\tilde{\pi}}(s) = \sum_{t=0}^{\infty} \gamma^t \Pr(s_t = s \mid s_0, \tilde{\pi})$.

For the adversarial nature player corresponding to agent $i$, the policy gradient with respect to the parameters $\omega_i$ and $\xi_i$ are given by:

$$\nabla_{\omega_i} J_i(\omega) = \mathbb{E}_{s,a \sim \zeta_{\tilde{\pi}_\theta}} \left[ \frac{\nabla_{\omega_i} \delta_{\omega_i}(a_i \mid s)}{\pi_{\theta_i}(a_i \mid s) + \delta_{\omega_i}(a_i \mid s)} \cdot Q_{\psi_i}(s, a) \right]. \tag{18}$$

$$\nabla_{\xi_i} J_i(\xi) \approx \mathbb{E}_{\tau \sim (P^H_{\xi_i}, \bar{\pi}_\theta)} \left[ \left( \sum_{t=0}^{T-1} \nabla_{\xi_i} \log P^H_{\xi_i}(s^{t+1} \mid s^t, a^t, \delta^t) \right) J_i(\pi, \tau) \right]. \tag{19}$$

Here, $J_i$ is the return for agent $i$, and $\tau$ represents the sampled trajectories. The detailed derivation is listed in B.5.

### 3.4.3 Actor-Critic Algorithm

As derived in the Eq. 16, the robust Q-function characterizes the expected return under the joint policy and uncertainty perturbations. In practice, we approximate this Q-function using a parameterized critic network to estimates the robust action-value function for each agent. Specifically, for agent $i$, the robust Q-function is represented as $Q_{\psi_i} : \mathcal{S} \times \mathcal{A} \to \mathbb{R}$, where $\psi_i \in \mathbb{R}^d$ denotes the critic parameters. The critic is updated through temporal-difference learning (Sutton, 1988; Konda & Tsitsiklis, 1999), which provides an online estimate of the robust return under the joint policy.

**Critic:**

$$\epsilon_i^t = R_i(s^t, a^t) + \gamma Q_{\psi_i^t}(s^{t+1}, a^{t+1}) - Q_{\psi_i^t}(s^t, a^t), \tag{20}$$

$$\psi_i^{t+1} = \psi_i^t + \alpha \cdot \epsilon_i^t \cdot \nabla_{\psi_i} Q_{\psi_i^t}(s^t, a^t), \tag{21}$$

where $\epsilon_i^t$ is the TD error for agent $i$, and $\alpha > 0$ is the learning rate for the critic.

With the critic providing an estimate of the robust Q-function, the actor updates the policy parameters by following the policy gradient derived in Eq 17.

**Actor:**

$$\theta_i^{t+1} = \theta_i^t + \beta \cdot \frac{\pi_{\theta_i^t}(a_i^t \mid s^t)}{\pi_{\theta_i^t}(a_i^t \mid s^t) + \delta_{w_i^t}(a_i^t \mid s^t)} \cdot \nabla_{\theta_i} \log \pi_{\theta_i^t}(a_i^t \mid s^t) \bar{Q}_{\psi_i^t}(s^t, a^t), \tag{22}$$

where $\beta > 0$ is the learning rate for the actor.

### 3.5 Implementation Details

In this section, we provide the implementation details for our proposed framework. Computing these entropy terms exactly in Eq. 6 is infeasible in practice due to the high-dimensional state and joint-action spaces in MARL. We therefore introduce practical estimation methods for environmental uncertainty and interaction uncertainty, which serve as the basis for constructing the corresponding uncertainty sets used during training.

To obtain a tractable estimate of the transition entropy, we assume that the environmental transition noise follows a Gaussian distribution. Under this assumption, the differential entropy admits the closed-form expression:

$$H(S' \mid s, a) = \frac{1}{2} \log \left( (2\pi e)^d \cdot |\sigma^2| \right), \tag{23}$$

 where $d$ is the dimensionality of the state space, and $\sigma^2$ is the predicted variance of the next-state distribution, to estimate the transition variance $\sigma$, we utilize a probabilistic neural network Specht (1990); Mohebali et al. (2020) to model the environmental dynamics. This network predicts both the mean and variance of the transition dynamics, enabling us to model the stochasticity in the environment effectively. The probabilistic dynamics model is trained by minimizing the negative log-likelihood loss that captures the error in both the predicted mean and variance of the transitions:

$$\mathcal{L}_{\text{dynamics}} = \frac{1}{N} \sum_{k=1}^{N} \left[ \frac{\| s'^k - \mu(s^k, a^k) \|^2}{2\sigma^2(s^k, a^k)} + \log \sigma(s^k, a^k) \right], \tag{24}$$

where $\mu(s^k, a^k)$ is the predicted mean, $\sigma(s^k, a^k)$ is the predicted variance, and $N$ is the number of samples in the batch.

Interaction uncertainty is quantified by the entropy of the joint action distribution $H(a \mid s)$, defined as

$$H(a \mid s) = -\sum_{a \in \mathcal{A}} \pi(a \mid s) \log \pi(a \mid s), \tag{25}$$

where $\pi(a \mid s)$ denotes the probability of executing joint action $a$ in state $s$. Since the joint action space grows exponentially with the number of agents, directly computing $\pi(a \mid s)$ is intractable. Following the decentralized Markov decision process framework (Oliehoek et al., 2016), we factorize the joint policy as

$$\pi(a \mid s) = \prod_{i=1}^{N} \pi_i(a_i \mid o_i), \tag{26}$$

where $o_i$ is the local observation of agent $i$. This factorization enables decentralized computation of the joint distribution from individual agent policies. To further reduce complexity, we approximate the entropy via Monte Carlo sampling: given $K$ sampled joint actions $\{a^k\}_{k=1}^{K}$ drawn from the factorized distribution, we estimate

$$H(a \mid s) \approx -\frac{1}{K} \sum_{k=1}^{K} \log \prod_{i=1}^{N} \pi_i(a_i^k \mid o_i). \tag{27}$$

This sampling-based estimation efficiently measures interaction uncertainty without enumerating the entire joint action space. Further implementation details are listed at Algorithm 1.

## 4 Experiments

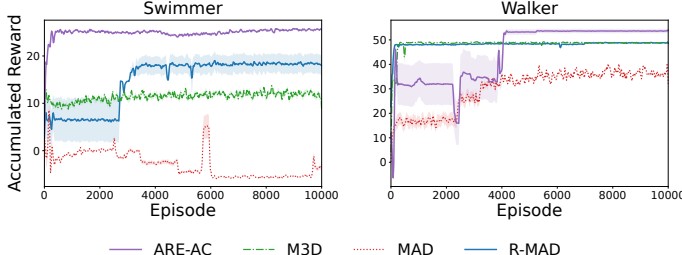

Figure 2: Accumulated reward under mid-level noise in multi-agent MuJoCo tasks.

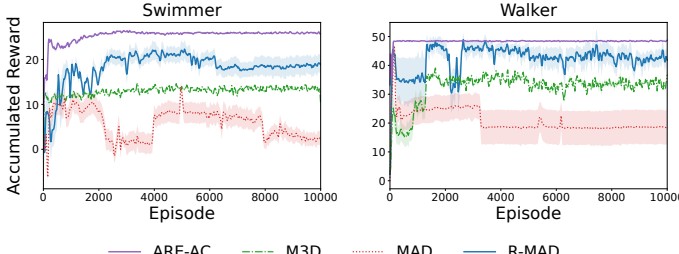

Figure 3: Accumulated reward under high-level noise in multi-agent MuJoCo tasks.

In this section, we conduct empirical evaluations to demonstrate the validity and effectiveness of the proposed method. Specifically, we aim to answer the following questions: **Q1. Performance Evaluation:** Can the proposed ARE framework ensure robust performance under different uncertainty scenarios? **Q2. Ablation Study:** How do different types of uncertainty and varying noise magnitudes affect policy performance? In particular, can the hierarchical uncertainty sets lead to less conservative performance? **Q3. Uncertainty Decomposition Analysis:** Can the ARE framework accurately tackle different sources of uncertainty? **Q4. Impact of Noise Levels and Settings:** How do varying noise levels and different noise settings influence policy robustness?

### 4.1 Environment Settings and Baselines

We evaluate our approach across benchmark environments from both the MPE environments (Lowe et al., 2017): cooperative navigation, keep-away, physical deception, predator-prey, and the multi-agent MuJoCo tasks (Peng et al., 2021): Cheetah $2 \times 3$, Cheetah $6 \times 1$, Swimmer, Walker. Detailed environment configurations are provided in Appendix C.1. To simulate different levels of uncertainty, we adjust $\rho_{\text{int}}$ and

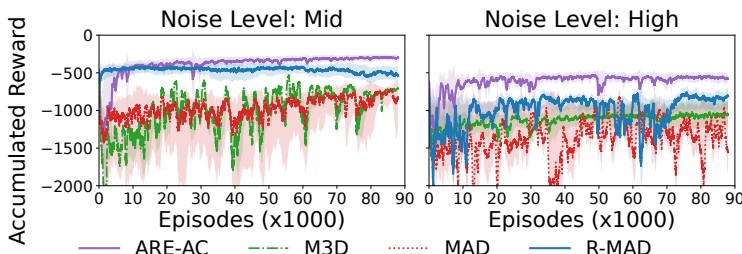

Figure 4: Cumulative reward in Cooperative Navigation under mid and high-level noise.

$\rho_{\text{env}}$ in Eq. 7 and Eq. 8, which control interaction and environmental uncertainty, respectively. These parameters do not directly set a fixed additive-noise scale; instead, they specify the radii of entropy-bounded uncertainty sets, so larger $\rho_{\text{int}}$ or $\rho_{\text{env}}$ permits more adversarial deviations and therefore induces stronger interaction/environmental noise. The corresponding noise levels, reward structures, and evaluation metrics are summarized in Appendix C.2.

Building on the actor–critic framework, we propose the **ARE-AC** algorithm, designed to achieve robust performance under both types of aleatoric uncertainty. We compare against three baselines. **M3DDPG (M3D)** extends MADDPG with a max-min formulation but only accounts for perturbations in agent policies (Li et al., 2019). **R-MADDPG (R-MAD)** introduces robustness against perturbations in the reward and transition models, but does not explicitly capture interaction uncertainty (Zhang et al., 2020b). Finally, **MADDPG (MAD)** serves as the standard multi-agent actor–critic baseline without robustness mechanisms (Lowe et al., 2017). For all baselines, we retain their original module designs as in the respective papers. However, to ensure fair comparison, we evaluate them under the same setting as ARE-AC, where both interaction and environmental noise are simultaneously present. In this way, although some baselines are inherently restricted to handling only one type of perturbation, our experiments highlight the advantage of ARE-AC in addressing multiple sources of aleatoric uncertainty within a unified framework.

## 4.2 Performance Evaluation

| Algorithm | Keep-Away | | Physical Deception | | Predator-Prey | | Multiwalker | |
|---|---|---|---|---|---|---|---|---|
| | Mid | High | Mid | High | Mid | High | Mid | High |
| ARE-AC | **2.32 ± 0.79** | **1.68 ± 0.98** | **1.34 ± 0.22** | **0.77 ± 0.11** | **-0.74 ± 0.18** | **-0.86 ± 0.21** | **0.12 ± 0.38** | **-1.65 ± 1.12** |
| R-MAD | 1.67 ± 0.88 | 1.07 ± 0.63 | 1.17 ± 0.30 | 0.52 ± 0.34 | -0.92 ± 0.26 | -1.13 ± 0.44 | -0.39 ± 0.24 | -2.97 ± 0.34 |
| M3D | 1.43 ± 0.72 | 0.84 ± 0.45 | 0.83 ± 0.17 | 0.27 ± 0.09 | -0.96 ± 0.30 | -1.69 ± 0.47 | -0.62 ± 0.29 | -3.31 ± 0.58 |
| MAD | 1.19 ± 0.62 | 0.39 ± 0.07 | 0.67 ± 0.14 | 0.17 ± 0.05 | -0.83 ± 0.21 | -1.96 ± 0.32 | -1.09 ± 0.47 | -5.28 ± 0.61 |

Table 1: Average return (± std) under **mid** and **high** noise levels across four environments.

We first evaluate all methods in the **Cooperative Navigation** environment under mid- and high-noise conditions (Fig. 4). **ARE-AC** consistently outperforms all baselines, exhibiting faster convergence, lower variance, and higher cumulative rewards. This superior performance arises from its ability to handle both *environmental* and *interaction* uncertainties simultaneously. In contrast, **M3D**, which focuses solely on policy-level perturbations, struggles to adapt when environmental transitions become stochastic. Conversely, **R-MAD** accounts only for environmental uncertainty and thus suffers from instabilities caused by interaction randomness among agents. **MAD** lacks any explicit robustness mechanism and performs the worst across all noise levels, highlighting the need for structured uncertainty modeling. We omit detailed comparisons under low noise conditions, as the perturbations are too small to significantly impact policy accuracy, resulting in comparable performance across all methods.

Beyond Cooperative Navigation, we further assess performance in the **Keep-Away**, **Physical Deception**, and **Predator–Prey** scenarios, as well as in the multi-agent MuJoCo benchmarks (Table 1, Fig. 2, and Fig. 3). Across these diverse domains, **ARE-AC** consistently achieves the best performance under both mid- and high-level noise, maintaining stable returns even when other methods degrade sharply. These results confirm that **ARE-AC** effectively unifies robustness against both environmental and interaction uncertainties, making it the most reliable framework across varying noise intensities.

### 4.3 Ablation Study

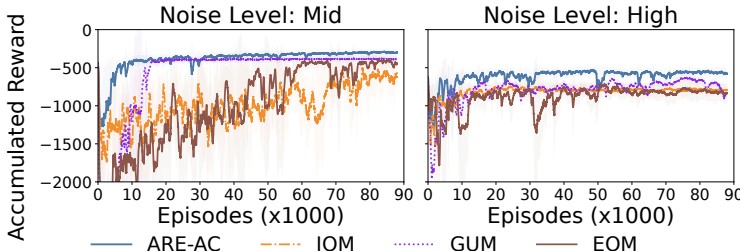

Figure 5: Ablation study of average return in Cooperative Navigation under mid and high-level noise.

Fig. 5 presents an ablation study comparing **ARE-AC** with three ablated variants under mid- and high-level noise. In these experiments, we explicitly control the construction of uncertainty sets to isolate the contributions of different components. The **General Uncertainty Model (GUM)** constructs a universal uncertainty set by jointly optimizing over environmental and interaction disturbances, but lacks the proposed hierarchical decomposition. The **Environmental-Only Model (EOM)** retains only the environmental uncertainty set and ignores interaction-level disturbances, while the **Interaction-Only Model (IOM)** preserves only the interaction uncertainty set and discards environmental disturbances. It is noted that ARE-AC consistently outperforms all variants under both mid- and high-level noise conditions. Compared to GUM, ARE-AC's superior performance validates the benefit of the proposed hierarchical optimization, which avoids the overly conservative policies that emerge when both uncertainties are maximized simultaneously. EOM and IOM, while still formulated within our unified ARE framework, exhibit significant degradation once either type of uncertainty is excluded. This result highlights that both environmental and interaction disturbances constitute critical sources of performance degradation, and omitting either one leads to policies that fail to maintain robustness under realistic noisy conditions.

### 4.4 Uncertainty Decomposition Analysis

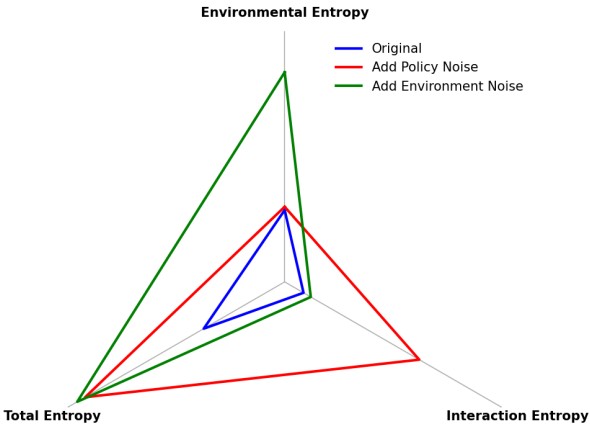

Figure 6: Radar plot of uncertainty decomposition.

To further verify whether our framework can accurately distinguish and capture the effects of environmental and interaction uncertainty introduced in Sec. 3.5, we conduct controlled experiments that selectively inject noise into different sources of uncertainty.

Specifically, we independently introduce high-level noise to either the environment (environmental noise) or the policy (interaction noise) and observe the resulting changes in entropy. As shown in Fig. 6, adding policy noise results in a significant increase in interaction entropy (Eq. 27) compared with original entropy

(blue line), while leaving environmental entropy (Eq. 23) largely unchanged. Conversely, when environmental noise is applied, environmental entropy rises sharply, whereas interaction entropy exhibits minimal variation. This clear separation demonstrates that each entropy measure is primarily influenced by its corresponding noise source, confirming the accuracy of our decomposition. We further examine sensitivity by gradually

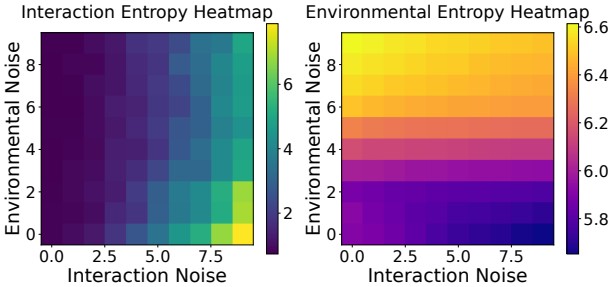

Figure 7: Heatmap visualization of uncertainty decomposition.

increasing the noise magnitude for both components. Fig. 7 illustrates the resulting trends of interaction and environmental entropy. Interaction entropy consistently increases with interaction noise while remaining stable under environmental noise, indicating that the entropy estimator in Eq. 27 is selectively responsive to policy-level perturbations. In contrast, environmental entropy increases smoothly with environmental noise while remaining insensitive to policy perturbations, demonstrating that the estimator defined in Eq. 23 accurately captures transition-level stochasticity. Together, these results validate the effectiveness of our entropy-based quantification in disentangling and measuring the two sources of aleatoric uncertainty.

## 4.5 Analysis of Uncertainty Types and Noise Scaling Effects

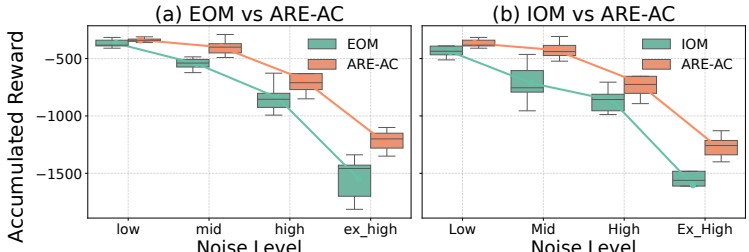

Figure 8: Performance under increasing environmental and interaction noise.

Fig. 8 compare the performance of our full uncertainty decomposition method (ARE-AC) against two partial modeling baselines: EOM (environment-only) and IOM (interaction-only). In Fig. 8a, we gradually increase the environmental noise while keeping the interaction noise fixed. ARE-AC maintains stable performance even under extremely high disturbance levels, whereas EOM suffers a sharp degradation once the environmental variance exceeds the level captured by its single uncertainty set. Conversely, Fig. 8b varies the interaction noise, where IOM fails to compensate for the adaptive behaviors of other agents, while ARE-AC continues to yield higher and more stable returns. These results demonstrate that ARE-AC can adaptively emphasize the dominant uncertainty source in each scenario and preserve robustness.

## 5 Conclusion

In this work, we proposed a robust Markov game framework that explicitly models two fundamental sources of uncertainty: *environmental* and *interaction* uncertainty. By introducing hierarchical entropy-based uncertainty sets, our framework captures these uncertainties in a unified yet structured manner, allowing agents to optimize against worst-case scenarios without incurring excessive conservatism. To attain the ARE, we developed tailored actor-critic algorithms and provided theoretical guarantees of their convergence and feasibility. Comprehensive experiments across both MPE and MuJoCo multi-agent benchmarks demonstrate that our method consistently outperforms existing baselines under various noise conditions.

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

# A    Notation Summary

To improve clarity and avoid potential confusion caused by the large number of symbols used throughout the paper, we summarize the key notations in Table 2. These notations are used consistently across the problem formulation, theoretical analysis, and algorithmic derivations.

| Symbol | Description |
| --- | --- |
| $\mathcal{S}$ | State space of the Markov game |
| $N$ | Number of agents |
| $\mathcal{A}_i$ | Action space of agent $i$ |
| $\mathcal{A}$ | Joint action space, $mathcalA = mathcalA_1 \times \cdots \times mathcalA_N$ |
| $\pi_i(a_i|s)$ | Policy of agent $i$ |
| $\pi = (\pi_1, \ldots, \pi_N)$ | Joint policy of all agents |
| $\delta_i$ | Policy perturbation representing interaction uncertainty |
| $\bar{\pi}_i$ | Perturbed policy of agent $i$, i.e., $\bar{\pi}_i = \pi_i + \delta_i$ |
| $\Delta_{\pi_i}$ | Interaction uncertainty set for policy perturbations |
| $\tilde{\pi}$ | Joint strategy over agents' policies and nature players' uncertainty |
| $P(s'|s,a)$ | Nominal transition dynamics |
| $\bar{P}(s'|s,a)$ | Perturbed transition kernel |
| $\bar{P}^H(s'|s,a,\delta)$ | Perturbed hierarchical transition kernel |
| $\tilde{P}_s$ | Environmental uncertainty set of transition kernels |
| $R_i(s,a)$ | Reward function of agent $i$ |
| $V_i(s)$ | Value function of agent $i$ |
| $\bar{V}_i(s)$ | Robust value function under uncertainty |
| $Q_i(s,a)$ | Robust action-value function for agent $i$ |
| $U_H$ | Hierarchical uncertainty set |
| $U_G$ | General (product) uncertainty set |
| $\rho_{\text{int}}$ | Interaction entropy bound |
| $\rho_{\text{env}}$ | Environmental entropy bound |
| $\gamma$ | Discount factor |
| $\eta$ | Policy of the nature player, representing the strategy over policy perturbation $\delta$ and transition $P$ |
| $\theta_i$ | Parameters of the policy $\pi_i$ for agent $i$ |
| $w_i$ | Parameters of the policy perturbation function $\delta_i$ (interaction uncertainty adversary) |
| $\xi_i$ | Parameters of the adversarial transition model $P^H_{\xi_i}$ (environmental uncertainty adversary) |
| $\psi_i$ | Parameters of the critic network approximating the robust Q-function $Q_{\psi_i}(s,a)$ |
| $\zeta_{\tilde{\pi}}$ | Discounted state visitation distribution induced by the joint strategy $\tilde{\pi}$ |
| $\epsilon^t_i$ | Temporal-difference (TD) error of agent $i$ at time step $t$ |
| $\alpha$ | Learning rate of the critic update |
| $\beta$ | Learning rate of the actor update |
| $\mu(s,a)$ | Predicted mean of the probabilistic transition model for the next state |
| $\sigma(s,a)$ | Predicted variance of the probabilistic transition model for the next state |

Table 2: Summary of key notations used in the paper.

# B    Mathematical Proof

## B.1    Proof of Theorem 3.2

*Proof.* To prove the existence and uniqueness of $\bar{V}_i^*(s)$, we rely on the Banach fixed-point theorem (Kardeş et al., 2011). The max-min Bellman operator $T$ is a contraction mapping, as established in (He et al., 2023). Specifically, for any two value functions $\bar{V}_1, \bar{V}_2 \in \mathcal{V}$, the operator $T$ satisfies:

$$\|T[\bar{V}_1] - T[\bar{V}_2]\|_\infty \leq \gamma \|\bar{V}_1 - \bar{V}_2\|_\infty.$$

where $\gamma \in [0,1)$ is the discount factor. By the Banach fixed-point theorem, $T$ has a unique fixed point $\bar{V}^* \in \mathcal{V}$, which satisfies $T[\bar{V}_*](s) = \bar{V}^*(s)$ for all $s \in \mathcal{S}$.

Thus, the robust value function $\bar{V}_i^*(s)$ exists and is unique. Moreover, since the Bellman operator $T$ is a contraction mapping, the value iteration sequence $\bar{V}_{k+1} = T[\bar{V}_k]$ converges to this unique fixed point from any initialization. □

## B.2 Proof of Lemma 3.3

*Proof.* Let $\mathcal{U}_G$ denote the general uncertainty set:

$$\mathcal{U}_G = \big\{ (\bar{P}, \delta) \mid \delta \in \Delta_\pi, \ \bar{P}(s' \mid s, a) \in \tilde{\mathcal{P}}_s(\pi), \forall s, s' \in \mathcal{S}, \forall a \sim \pi \big\}.$$

The hierarchical entropy-based uncertainty set $\mathcal{U}_H$ is defined as:

$$\mathcal{U}_H = \big\{ (\bar{P}^H, \delta) \mid \delta \in \Delta_\pi, \ \bar{P}^H(s' \mid s, \bar{a}, \delta) \in \tilde{\mathcal{P}}_s(\bar{\pi}), \forall s, s' \in \mathcal{S}, \forall \bar{a} \sim \bar{\pi} \big\},$$

Both $\mathcal{U}_G$ and $\mathcal{U}_H$ share the same set for $\delta$, i.e., $\delta \in \Delta_\pi$. Thus, the $\delta$ component of $\mathcal{U}_H$ is a subset of that in $\mathcal{U}_G$.

In $\mathcal{U}_H$, $\bar{P}^H$ is conditionally dependent on $\delta$, and $\bar{P}^H \in \tilde{\mathcal{P}}_s(\bar{\pi})$. Since $\tilde{\mathcal{P}}_s(\bar{\pi}) \subseteq \tilde{\mathcal{P}}_s(\pi)$, we have:

$$\bar{P}^H \in \tilde{\mathcal{P}}_s(\bar{\pi}) \implies \bar{P}^H \in \tilde{\mathcal{P}}_s(\pi).$$

Since both the components in $U_H$ are subsets of the $U_G$, $U_H$ is a subset of the $U_G$. □

## B.3 Proof of Theorem 3.4

*Proof.* To prove that the performance lower bound under the hierarchical entropy-based uncertainty set is no worse than that under the general uncertainty set, we proceed as follows:

$$\inf_{(\bar{P}^H, \delta) \in \mathcal{U}_H} V_{\bar{P}^H, \delta}(s) = \inf_{\substack{\bar{P}^H \in \tilde{\mathcal{P}}_s(\bar{\pi}), \\ \delta \in \Delta_\pi}} \mathbb{E}\left[ \sum_{t=0}^\infty \gamma^t R(s_t, a_t) \mid s_0 = s \right]$$

$$\geq \inf_{\substack{\bar{P} \in \tilde{\mathcal{P}}_s(\pi), \\ \delta \in \Delta_\pi}} \mathbb{E}\left[ \sum_{t=0}^\infty \gamma^t R(s_t, a_t) \mid s_0 = s \right]$$

$$= \inf_{(\bar{P}, \delta) \in \mathcal{U}_G} V_{\bar{P}, \delta}(s).$$

The inequality holds because $\tilde{\mathcal{P}}_s(\bar{\pi}) \subseteq \tilde{\mathcal{P}}_s(\pi)$ is proved by Lemma 3.3 and the works (Kardeş et al., 2011; Xu et al., 2023) demonstrated, which means that the hierarchical entropy-based uncertainty set imposes additional constraints on $\mathcal{P}$ compared to the general uncertainty set. Thus, we conclude:

$$\inf_{(\bar{P}^H, \delta) \in \mathcal{U}_H} V_{\bar{P}^H, \delta}(s) \geq \inf_{(\bar{P}, \delta) \in \mathcal{U}_G} V_{\bar{P}, \delta}(s).$$

□

## B.4 Proof of Theorem 3.6

*Proof.* Following the proof (Zhang et al., 2020b; Kardeş et al., 2011), we establish the existence of an ARE by applying Kakutani's Fixed Point Theorem.

Define the correspondence $\Phi_i : \Pi \to 2^\Pi$, where $\Pi$ denotes the space of agent $i$'s policies, as follows:

$$\Phi_i(\pi) = \Bigg\{ \pi_i^* : \ \pi_i(\cdot \mid s) \in \arg\max_{\pi_i(\cdot \mid s)} \min_{\substack{\delta_i \in \Delta_{\pi_i} \\ \bar{P}^H(\cdot \mid s, a, \delta) \in \tilde{\mathcal{P}}_s(\bar{\pi})}} \sum_{a_i \in \mathcal{A}_i} \pi_i(a_i \mid s)$$

$$\prod_{j \neq i} \bar{\pi}_j^*(a_j \mid s, \eta_j^*) \Big( R_i(s, a) + \gamma \sum_{s' \in \mathcal{S}} \bar{P}^H(s' \mid s, a, \delta) \bar{V}_i(s') \Big) \Bigg\}.$$

where $\bar{V}_i(s)$ has been proved to converge to the unique $\bar{V}_i^*(s)$. To apply Kakutani's Fixed Point Theorem, we verify the following conditions:

- **Non-emptiness:** For any joint policy $\pi \in \Pi$, the maximization over $\pi_i$ and minimization over $\delta$ and $\tilde{\mathcal{P}}_s(\pi)$ are well-defined because the feasible sets $\Delta_\pi$ and $\tilde{\mathcal{P}}_s(\pi)$ are compact and convex. Thus, the solution set of the optimization problem, and hence $\Phi_i(\pi)$, is non-empty.

- **Compactness and Convexity:** The set $\Phi_i(\pi)$ is compact and convex. Compactness arises because the feasible sets $\Delta_\pi$ and $\tilde{\mathcal{P}}_s(\pi)$ are compact, and the objective function is continuous in $\pi_i$.

  To show that $\Phi_i(\pi)$ is convex, we consider each agent's best-response problem. Fix the policies of the other agents $\pi_j$, and select two arbitrary policies $\pi_i^1, \pi_i^2 \in \Phi_i(\pi)$ for agent $i$. For any $\lambda \in [0, 1]$, define $\pi_i^\lambda = \lambda \pi_i^1 + (1 - \lambda)\pi_i^2$. Given the structure of the policy function $\pi$, we have:

  $$\pi(a \mid s) = \prod_{i=1}^{N} \pi_i(a_i \mid s) = \pi_i(a_i \mid s) \prod_{j \neq i} \pi_j(a_j \mid s).$$

  which is affine with respect to $\pi_i(a_i \mid s)$ when $\pi_j$ is fixed. Since $\pi_i(a_i \mid s)$ is linear in $\pi_i$, any convex combination of $\pi_i^1$ and $\pi_i^2$ satisfies:

  $$\pi_i^\lambda(a_i \mid s) = \lambda \pi_i^1(\cdot \mid s) + (1 - \lambda)\pi_i^2(\cdot \mid s).$$

  Furthermore, the feasible sets $\Delta_\pi$ and $\tilde{\mathcal{P}}_s(\pi)$ are convex by assumption, leveraging the fact that Shannon entropy constraints define convex level sets. Therefore, $\Phi_i(\pi)$ is a convex correspondence.

- **Compactness of the Policy Space $\Pi$:** The policy space $\Pi$ is compact because $\mathcal{S}$ and $\mathcal{A}$ are finite, and the joint policies $\pi$ are probability distributions over $\mathcal{A}$. This ensures that $\Pi$ is a subset of a finite-dimensional simplex, which is compact.

By Kakutani's Fixed Point Theorem, the correspondence $\Phi_i$ admits at least one fixed point. This fixed point corresponds to a policy $\pi_i^*$ satisfying the equilibrium condition. $\square$

## B.5 Proof of Policy Gradient Updating Rules

*Proof.* We derive the policy gradient update for each agent $i \in \mathcal{N}$ under the defined joint strategy $\tilde{\pi}_\theta = (\pi_{\theta_1}, \delta_{w_1}, \bar{P}_{\xi_1}^H, \ldots, \pi_{\theta_N}, \delta_{w_N}, \bar{P}_{\xi_N}^H)$.

Taking the gradient of $J_i(\theta)$ with respect to $\theta_i$, we start by expanding:

$$\nabla_{\theta_i} J_i(\theta) = \nabla_{\theta_i} \bar{V}_{\tilde{\pi}_\theta}^i(s_0).$$

Using the recursive definition of

$$J_i(\theta) = \bar{V}^i_{\tilde{\pi}_\theta}(s_0) \ = \ \sum_{s \in \mathcal{S}} \sum_{a \in \mathcal{A}} \zeta_{\tilde{\pi}_\theta}(s,a) \, Q_{\psi_i}(s,a)$$

$$\nabla_{\theta_i} J_i(\theta) = \nabla_{\theta_i} \sum_{s,a} \zeta_{\tilde{\pi}_\theta}(s,a) \, Q_{\psi_i}(s,a)$$

$$= \sum_{s,a} \nabla_{\theta_i} \Big[ \zeta_{\tilde{\pi}_\theta}(s,a) \Big] \, Q_{\psi_i}(s,a)$$

$$= \sum_{s,a} \zeta_{\tilde{\pi}_\theta}(s,a) \, \nabla_{\theta_i} \log \Big[ \zeta_{\tilde{\pi}_\theta}(s,a) \Big] \, Q_{\psi_i}(s,a)$$

$$= \sum_{s,a} \zeta_{\tilde{\pi}_\theta}(s,a) \, \nabla_{\theta_i} \log \Big[ \tilde{\pi}_\theta(a \mid s) \Big] \, Q_{\psi_i}(s,a),$$

$$= \sum_{s,a} \zeta_{\tilde{\pi}_\theta}(s,a) \, \nabla_{\theta_i} \log \Big[ \pi_{\theta_i}(a_i \mid s) + \delta_{w_i}(a_i \mid s) \Big] \, Q_{\psi_i}(s,a)$$

$$= \sum_{s,a} \zeta_{\tilde{\pi}_\theta}(s,a) \, \frac{\nabla_{\theta_i} \pi_{\theta_i}(a_i \mid s)}{\pi_{\theta_i}(a_i \mid s) + \delta_{w_i}(a_i \mid s)} \, Q_{\psi_i}(s,a)$$

$$= \sum_{s,a} \zeta_{\tilde{\pi}_\theta}(s,a) \, \frac{\pi_{\theta_i}(a_i \mid s)}{\pi_{\theta_i}(a_i \mid s) + \delta_{w_i}(a_i \mid s)} \nabla_{\theta_i} \log \pi_{\theta_i}(a_i \mid s) \, Q_{\psi_i}(s,a)$$

$$= \mathbb{E}_{s,a \sim \zeta_{\tilde{\pi}_\theta}} \left[ \frac{\pi_{\theta_i}(a_i \mid s)}{\pi_{\theta_i}(a_i \mid s) + \delta_{w_i}(a_i \mid s)} \, \nabla_{\theta_i} \log \pi_{\theta_i}(a_i \mid s) \, Q_{\psi_i}(s,a) \right].$$

Similarly, we derive the gradient update of the $\omega$:

$$\nabla_{\omega_i} J_i(\omega) = \ \nabla_{\omega_i} \sum_{s,a} \zeta_{\tilde{\pi}_\theta}(s,a) \, Q_{\psi_i}(s,a)$$

$$= \ \sum_{s,a} \zeta_{\tilde{\pi}_\theta}(s,a) \, \nabla_{\omega_i} \log \Big[ \tilde{\pi}_\theta(a \mid s) \Big] \, Q_{\psi_i}(s,a)$$

$$= \ \sum_{s,a} \zeta_{\tilde{\pi}_\theta}(s,a) \, \frac{\nabla_{\omega_i} \delta_{\omega_i}(a_i \mid s)}{\pi_{\theta_i}(a_i \mid s) + \delta_{\omega_i}(a_i \mid s)} \, Q_{\psi_i}(s,a)$$

$$= \ \mathbb{E}_{s,a \sim \zeta_{\tilde{\pi}_\theta}} \left[ \frac{\nabla_{\omega_i} \delta_{\omega_i}(a_i \mid s)}{\pi_{\theta_i}(a_i \mid s) + \delta_{\omega_i}(a_i \mid s)} \, Q_{\psi_i}(s,a) \right].$$

Additionally, we use sampling-based method to update $\xi$:

$$J_i(\xi) = \sum_{s \in \mathcal{S}} \sum_{a \in \mathcal{A}} \zeta_{\tilde{\pi}_\theta}(s,a) \, Q_{\psi_i}(s,a),$$

$$\tau = (s_0, a_0, s_1, a_1, \ldots, s_{T-1}, a_{T-1}, s_T),$$

$$\nabla_{\xi_i} \log \Pr_{\xi_i}(\tau) = \sum_{t=0}^{T-1} \nabla_{\xi_i} \log P^H_{\xi_i}\big(s^{t+1} \mid s^t, a^t\big),$$

$$\nabla_{\xi_i} J_i(\xi) \approx \mathbb{E}_{\tau \sim (P^H_{\xi_i}, \tilde{\pi}_\theta)} \left[ \left( \sum_{t=0}^{T-1} \nabla_{\xi_i} \log P^H_{\xi_i}\big(s^{t+1} \mid s^t, a^t\big) \right) \cdot J_i(s,\tau) \right].$$

$\square$

## B.6 Convergence of Hierarchical Robust Q-value

*Proof.* Consider the hierarchical robust Bellman operator $\mathcal{T}^H$ acting on a Q-function $Q_i \in \mathcal{Q}$:

$$(\mathcal{T}^H Q_i)(s,a) = R_i(s,a) + \gamma \min_{\substack{\delta_i \in \Delta_{\pi_i} \\ \bar{P}^H \in \tilde{\mathcal{P}}_s}} \mathbb{E}_{s' \sim \bar{P}^H(\cdot|s,a,\delta)}[\bar{V}_i(s')]$$

where $V_i(s') = \max_{a'} Q_i(s',a')$. Let $Q_i^k, Q_i^{k+1}$ be two arbitrary Q-functions. For a fixed state-action pair $(s,a)$, we have:

$$
\begin{aligned}
|(\mathcal{T}^H Q_i^k)(s,a) - (\mathcal{T}^H Q_i^{k+1})(s,a)| &= \gamma \left| \min_{\substack{\delta_i \in \Delta_{\pi_i} \\ \bar{P}^H \in \tilde{\mathcal{P}}_s}} \mathbb{E}_{\bar{P}^H}[\bar{V}_i^k(s')] - \min_{\substack{\delta_i \in \Delta_{\pi_i} \\ \bar{P}^H \in \tilde{\mathcal{P}}_s}} \mathbb{E}_{\bar{P}^H}[\bar{V}_i^{k+1}(s')] \right| \\
&\leq \gamma \max_{\substack{\delta_i \in \Delta_{\pi_i} \\ \bar{P}^H \in \tilde{\mathcal{P}}_s}} \left| \mathbb{E}_{\bar{P}^H}[\bar{V}_i^k(s') - \bar{V}_i^{k+1}(s')] \right| \\
&\leq \gamma \max_{\substack{\delta_i \in \Delta_{\pi_i} \\ \bar{P}^H \in \tilde{\mathcal{P}}_s}} \mathbb{E}_{\bar{P}^H} |[\bar{V}_i^k(s') - \bar{V}_i^{k+1}(s')]| \\
&\leq \gamma \|\bar{V}_i^k(s') - \bar{V}_i^{k+1}(s')\|_\infty
\end{aligned}
$$

Since

$$
\begin{aligned}
\|\bar{V}_i^k(s') - \bar{V}_i^{k+1}(s')\|_\infty &\leq \|\bar{V}_i^k(s') - \bar{V}_i^{k+1}(s')\|_1 \\
&= \sum_{s'} |\bar{V}_i^k(s') - \bar{V}_i^{k+1}(s')| \\
&\leq \sum_{s'} |\max_{a'} Q_i^k(s',a') - \max_{a'} Q_i^{k+1}(s',a')| \\
&\leq \sum_{s'} \max_{a'} |Q_i^k(s',a') - Q_i^{k+1}(s',a')| \\
&\leq \|Q_i^k - Q_i^{k+1}\|_\infty
\end{aligned}
$$

Then we have that:

$$|(\mathcal{T}^H Q_i^k)(s,a) - (\mathcal{T}^H Q_i^{k+1})(s,a)| \leq \gamma \|Q_i^k - Q_i^{k+1}\|_\infty$$

Taking the supremum over all $(s,a)$ on the left-hand side yields $\|\mathcal{T}^H Q_1 - \mathcal{T}^H Q_2\|_\infty \leq \gamma \|Q_1 - Q_2\|_\infty$. Since $0 \leq \gamma < 1$, the operator $\mathcal{T}^H$ is a $\gamma$-contraction mapping. By the Banach Fixed-Point Theorem, $Q$ converges to a unique $Q^*$. $\square$

# C Environment Settings

## C.1 Environment Details

### C.1.1 Cooperative Navigation

This scenario involves **3 agents** and **3 landmarks**. The agents receive rewards based on their distance to the landmarks, with the goal of minimizing the distance between themselves and any landmark. However, collisions between agents result in penalties. Therefore, the agents must learn to efficiently allocate themselves across the landmarks while avoiding collisions.

### C.1.2 Predator-Prey

In the predator-prey environment, there are **3 faster "good agents"** and **1 adversary** who is slower. The goal of the good agents is to evade capture, while the adversaries aim to catch them. The environment also includes large, immovable obstacles, which create barriers and add complexity to the movement strategies of the agents.

### C.1.3 Keep-Away

This scenario includes **1 agent**, **1 adversary**, and **1 landmark**. The agent earns rewards by staying close to the landmark, while the adversary gains rewards by either approaching the landmark or forcing the agent away from it. As a result, the adversary learns strategies to block or disrupt the agent's attempts to maintain proximity to the landmark.

### C.1.4 Physical Deception

In this environment, there are **2 good agents**, **1 adversary**, and **2 landmarks**. Among the landmarks, one is designated as the "target landmark." Good agents are rewarded for being close to the target landmark but are penalized if the adversary approaches it. Meanwhile, the adversary earns rewards for getting closer to the target landmark, despite not knowing which one it is. To succeed, the good agents must learn to coordinate and strategically split up, effectively misleading the adversary into targeting the wrong landmark.

### C.1.5 Multi-Agent MuJoCo Tasks

To evaluate robustness and scalability in continuous control settings, we further employ a set of **multi-agent MuJoCo benchmarks**. These tasks include **Cheetah** $2 \times 3$, **Cheetah** $6 \times 1$, **Swimmer**, and **Walker**. In the Cheetah environments, multiple limbs (or agents) must coordinate to generate efficient locomotion while maintaining stability under external perturbations. The **Cheetah** $2 \times 3$ setup partitions the body into two groups of three actuators each, promoting inter-group coordination, whereas the **Cheetah** $6 \times 1$ task treats all six actuators as independent agents that must achieve synchronized gait control. The **Swimmer** and **Walker** tasks evaluate robustness in low- and high-dimensional continuous action spaces, respectively, where agents must maintain smooth propulsion and balance under stochastic disturbances. These environments collectively provide diverse testing conditions for analyzing the robustness and coordination capabilities of our framework across both discrete and continuous domains.

## C.2 Metrics

We evaluate the robustness and effectiveness of the proposed framework under three uncertainty levels: **low**, **mid**, and **high**. The bounds for interaction and environmental uncertainties ($\rho_{\text{int}}$ and $\rho_{\text{env}}$) are set to 0.1, 0.5, and 1.5 for these three levels, respectively. Each experiment is repeated with five fixed random seeds (`111-120`) to ensure reproducibility, and results are averaged across these runs.

**Cumulative Reward.** The primary performance metric is the cumulative reward (or return), defined as the sum of all rewards obtained within one episode:

$$R_{\text{total}} = \sum_{t=0}^{T} R^t,$$

which measures the overall task success and long-term coordination ability of the agents.

**Reward Structure in MPE.** In the Multi-Agent Particle Environment (MPE) tasks, the reward for each agent is defined as the negative Euclidean distance to its target (e.g., landmark or goal position), encouraging agents to minimize their distance while avoiding penalties such as collisions. For example, in *Cooperative Navigation*, each agent receives $r_i = -\|p_i - p_{\text{landmark}}\|_2$ plus a small penalty when colliding with others. This design encourages both spatial efficiency and cooperative distribution across goals.

**Reward Structure in MuJoCo.** In the continuous-control MuJoCo tasks, the total reward at each step combines several components: a forward-progress term, a control-effort penalty, and a survival bonus. The instantaneous reward is given by

$$r_t = r_{\text{forward}} - c_{\text{control}} + r_{\text{survive}},$$

where $r_{\text{forward}}$ measures the agent's velocity in the target direction, $c_{\text{control}}$ penalizes large actuator torques or joint accelerations, and $r_{\text{survive}}$ provides a small constant bonus for maintaining a stable posture. This formulation encourages energy-efficient and stable locomotion under uncertainty.

## C.3 Uncertainty Injection and Baselines' Settings

In this subsection, we describe how environmental and interaction uncertainties are instantiated during training and evaluation, and how these uncertainty sources are applied consistently across all compared methods. Rather than directly prescribing the magnitude of injected noise, we quantify uncertainty by the sizes of entropy-bounded uncertainty sets, parameterized by $\rho_{\text{env}}$ and $\rho_{\text{int}}$. These uncertainty levels are chosen to align with the noise regimes commonly used in prior robust MARL baselines. In particular, several methods such as R-MAD (Zhang et al., 2020b) introduce transition uncertainty by directly perturbing the transition dynamics (e.g., $\bar{P} = P + \delta$), where larger perturbations correspond to larger uncertainty sets. To enable a fair comparison, we calibrate the entropy-based uncertainty budgets $\rho_{\text{env}}$ and $\rho_{\text{int}}$ so that they induce uncertainty levels comparable to the low, medium, and high regimes used in these baseline methods.

**Environmental Uncertainty** Environmental uncertainty models stochasticity in the transition dynamics. For each environment, we inject environmental noise by perturbing the nominal transition function $P(s' \mid s, a)$ with a stochastic disturbance $\rho_{\text{env}}$ in Eq. 8 that controls the noise magnitude. We define three noise regimes: We consider three environmental noise regimes, corresponding to $\rho_{\text{env}} = 0.1$ (low), $\rho_{\text{env}} = 0.5$ (mid), and $\rho_{\text{env}} = 1.5$ (high), respectively. For MuJoCo environments, the noise is applied to joint accelerations and velocities; for MPE tasks, the noise is used to agent positions and velocities after each transition.

**Interaction Uncertainty** Interaction uncertainty captures stochasticity and unpredictability in agents' action selections. At each timestep, the executed action $\tilde{a}_i$ for agent $i$ is sampled from a perturbed policy that $\rho_{\text{int}}$ in Eq. 7 controls the strength of interaction noise. We consider: $\rho_{\text{int}} = 0.1$ (low), $\rho_{\text{int}} = 0.5$ (mid), and $\rho_{\text{int}} = 1.5$ (high).

**Combined Uncertainty Setting** Unless otherwise stated, both environmental and interaction uncertainties are simultaneously present during evaluation. That is, at each timestep, agents execute perturbed actions $\tilde{a}_i$ and transitions evolve according to the noisy dynamics described above.

| Parameter | Description | Value |
|---|---|---|
| Episode Length | Length of each episode in training | 50 |
| Memory Length | Size of the replay buffer | $1 \times 10^5$ |
| Tau | Soft update coefficient for target networks | 0.001 |
| Gamma | Discount factor for future rewards | 0.95 |
| Seed | Seed for random number generators | 111 to 115 |
| Actor Learning Rate | Learning rate for actor-network | 0.001 |
| Critic Learning Rate | Learning rate for critic network | 0.0001 |
| Batch Size | Number of samples per batch update | 64 |
| Epsilon Decay | Rate of decay for exploration epsilon | 10000 |

Table 3: Detailed hyperparameters used in the experiment.

**Baseline-Specific Handling of Uncertainty** We now clarify how each baseline interacts with the injected uncertainties:

- **MADDPG (MAD)**: Does not explicitly model uncertainty. Both environmental and interaction noise are treated as exogenous disturbances.
- **M3DDPG (M3D)**: Models adversarial perturbations at the policy level and is therefore explicitly robust to interaction uncertainty. Environmental noise is injected externally and not optimized adversarially.
- **R-MADDPG (R-MAD)**: Models adversarial perturbations in transition and reward dynamics, and is therefore robust to environmental uncertainty. Interaction noise is injected externally and not optimized adversarially.
- **ARE-AC (Ours)**: Explicitly models both uncertainty sources using hierarchical entropy-based uncertainty sets. Interaction perturbations are optimized at the upper level, and environmental perturbations are optimized conditionally at the lower level.

## D  Hyperparameters

In this section, we introduce the detailed setting of our algorithm and experiments.

First of all, the hyperparameters are defined as:

Besides, our detailed neural network structure is listed as follows:

| Network Type | Input | Layers and Neurons | Activation Functions |
|---|---|---|---|
| Actor | Observation (`dim_observation`) | FC1: 500, FC2: 128, FC3: `dim_action` | ReLU, ReLU, Tanh |
| Critic | Observation and Action (`n_agent` × `dim_observation` + `n_agent` × `dim_action`) | FC1: 1024, FC2: 512, FC3: 300, FC4: 1 | ReLU, ReLU, ReLU, None |
| Adv_Actor_Type_1 | Observation (`dim_observation`) | FC1: 500, FC2: 128, FC3: 1 | ReLU, ReLU, Tanh |
| Adv_Actor_Type_2 | Extended Observation (`dim_observation` + 1) | FC1: 500, FC2: 128, FC3: 1 | ReLU, ReLU, Tanh |
| Adv_Critic | Observation and Action (`n_agent` × `dim_observation` + `n_agent` × `dim_action`) | FC1: 1024, FC2: 512, FC3: 300, FC4: 1 | ReLU, ReLU, ReLU, None |
| Transition Model | State and Action (`state_dim` + `action_dim`) | FC1: `hidden_dim`, FC2: `hidden_dim`, Mean and Logvar Heads: `state_dim` | ReLU, ReLU, None, None |

Table 4: Detailed Architecture of Neural Networks

## E  Algorithm Details

---
**Algorithm 1** Actor-Critic for Robust Multi-Agent RL with Uncertainty Decomposition (ARE-AC)

---
**Require:** Initialize critic network parameters $\{\psi_i\}_{i \in \mathcal{N}}$, adversarial parameters $\{\omega_i\}_{i \in \mathcal{N}}$, $\{\xi_i\}_{i \in \mathcal{N}}$, policy parameters $\{\theta_i\}_{i \in \mathcal{N}}$, probabilistic dynamics model $\phi$ and replay buffer $\mathcal{D}$.

1: **for** episode = 1 to $M$ **do**
2:   Receive an initial state $s_0$.
3:   **for** $t = 1, \ldots, T$ **do**
4:     **for** $i \in \mathcal{N}$ **do**
5:       Sample a set of candidate actions $\{a^k\}_{k=1}^K$.
6:       Compute entropy based on sampled actions, current states, and policies to determine uncertainty sets by Eq. 23 and 27.
7:       Obtain strategy perturbation $\delta_{\omega_i}(\cdot \mid s^t)$ based on current state.
8:       Obtain transition perturbation $\bar{P}_{\xi_i}^H(\cdot \mid s^t, a^t, \delta_{\omega_i})$ based on states and policy perturbation.
9:       Sample action $a_i \sim \pi_{\theta_i}(\cdot \mid s^t) + \delta_{\omega_i}(a_i \mid s^t)$.
10:     **end for**
11:     Execute action $a$, observe new state $s^{t+1}$ (perturbed by $\bar{P}_{\xi_i}^H$), and receive rewards $R^t$.
12:     Store $(s^t, a, \delta_{\omega_i}, \bar{P}_{\xi_i}^H, R^t, s^{t+1})$ in replay buffer $\mathcal{D}$.
13:     Update $s^t \leftarrow s^{t+1}$.
14:   **end for**
15:   **for** $i \in \mathcal{N}$ **do**
16:     Sample mini-batch $(s^t, a^t, \delta_{\omega_i}, \bar{P}_{\xi_i}^H, R^t, s^{t+1})_{t=1}^S$ from $\mathcal{D}$.
17:     Update critic by the Eq. 20.
18:     Update actor using the Eq. 22.
19:     Update adversarial strategy perturbation network $\omega_i$ by Eq. 18.
20:     Update adversarial transition perturbation network $\xi_i$ by Eq. 19.
21:     Update the probabilistic dynamics model by Eq. 24.
22:   **end for**
23: **end for**

---

