# OpenReview forum: "A Unified Framework with Environmental and Interaction Uncertainty for Robust Multi-Agent Reinforcement Learning"
_TMLR — Rejected by TMLR_

### Review · Reviewer_hmJZ · 2026-03-04

**Summary Of Contributions:**

This work tackles the problem of learning robust joint policies in MARL by modeling aleatoric noise as adversarial nature players. The key novelty is a decomposition of the aleatoric noise into environmental and interaction uncertainties, allowing to learn less conservative policies than previous robust MARL approaches.

Leveraging this decomposition, the paper develops new policy gradient and actor-critic algorithms, and demonstrates how they learn a new solution concept of an Aleatoric Robust Equilibrium (ARE).

Strengths:

- Robust MARL is a problem of great importance and the proposed solution is both theoretically motivated and empirically verified against existing robust MARL algorithms.
- The empirical validation seems sound

My primary source of confusion is the difference between this work and standard robust MARL approaches, both in the motivation and the approach.

- The motivation of the problem is not clear enough. While the conservatism of previous robust MARL methods is criticized, there is a lack of motivation on why their approach is not realistic. A concrete example or more intuition on why we need to integrate the causal dependence would be appreciated. Why should we expect the uncertainty on the transitions to be dependent on the policy uncertainty?
- Multiple typos or mistakes are present in the formulas, which prevented me to understand the definition of certain terms and the details of the proposed methods.

**Audience:**

Yes

**Audience Explanation:**

Yes, robust MARL is of interest to the community. Multi-agent reinforcement learning in general is of increasing importance, and there is a need for robust solutions given the uncertainty related to the modeling of other players’ strategies.

**Broader Impact Concerns:**

There are no ethical concerns associated with this work.

**Claims And Evidence:**

No

**Claims Explanation:**

The authors point out that previous robust MARL approaches produce overly conservative policies because of a simplified analysis of aleatoric uncertainty stemming from stochasticity in the environment dynamics and the other players’ behavior. This is the motivation for their new formulation treating the two aspects independently.

However, the current state of the paper makes it hard for me to correctly evaluate the proposed solution. A few derivations are not clear to me, I believe this is not necessary a flaw of the proposed method but an issue with the presentation.

**Requested Changes:**

Before providing my comments, I should mention that I am not an expert in robust RL. However, I wasn’t able to follow the main derivations of the work, and this is I believe because of multiple typos or mistakes throughout the work.

I therefore think that the paper writing could be significantly strengthened, and the motivation could be clearer. Specifically, I would like to see adjustments to the submission on the following points.

- I agree with the analysis that allowing the nature players to couple the uncertainty on other players’ policies and the environment produce robust policies that can often be overly conservative. However, the motivation for the proposed solution is not clear to me. What is the theoretical motivation to discard the possibility of this coupling? How is this avoided by the proposed method?
- Equations 2 and 3 are the same. The notation of the expected return is not consistent, it is here defined on both states and policies, and later only used with parametrizations. Similarly, making the dependence of the value function on pi explicit would make things clearer.
- There seems to be a typo on Equation 6, as $\overline{\mathcal{P}}_s$ is not used and $\tilde{\mathcal{P}}$ is used in the formula
- Equation 9 depends on the action $\tilde{a}$ which is mentioned to be sampled from the policy. How is the sampling procedure affecting the definition of the set? Is there an expectation missing somewhere?
- In the proof of Theorem 3.6, why are the transitions independent of delta? One idea of the paper is to condition the environment uncertainty to the policy uncertainty, as in the definition of $\mathcal{U}_H$. However, this doesn’t seem to be the case further in the paper, which makes me wonder what the difference is with the more standard robust formulation. Please also clarify the formula for $\pi(a | s)$.
- Equation 23: please clarify the implication of this approximation. It seems to me that you are implicitly assuming gaussian environmental noise.
- Equations 25-27: why are you using the notation P instead of the joint policy already introduced?
- The work mentions theoretical guarantees on the convergence of the actor-critic algorithms. Where are the convergence guarantees explicited in the work?

Another minor point is:

- Please improve the writing in Section 3.3.3 when introducing the critic and actor updates. Currently the introduction of both formulas are not well integrated, breaking the flow of the writing.

---

> ### Author Response · Authors · 2026-03-16
> **Reply [1/2]**
>
> We sincerely thank the reviewer for taking the time to carefully read our paper and for providing such valuable comments. We address each concern below with clarifications.
>
> ---
>
> ### Q1. What is the theoretical motivation to discard the possibility of this coupling? How is this avoided by the proposed method?
>
> **A1**
>
> Traditional robust MARL formulations typically adopt an independent **“max-of-both”** uncertainty structure, where the adversary aims to select the worst-case disturbances that degrade the expected return. Since larger uncertainty typically leads to lower return, this adversarial objective can be expressed as minimizing the value function over the admissible disturbance sets
>
> $\min_{\delta}\min_{\bar{P}} V$,
>
> where the adversary can simultaneously choose the worst policy perturbation and the worst environmental transition. This effectively corresponds to optimizing over the product of two uncertainty sets, which often leads to overly conservative policies.
>
> Our method instead introduces a **hierarchical dependency**
>
> $\min_{\delta}\min_{\bar{P}^H(\delta)} V$,
>
> where environmental disturbances are conditioned on the perturbed policy. This formulation avoids unrealistic combinations of disturbances and results in a strictly smaller uncertainty region compared to the independent product set.
>
> As the larger uncertainty set allows the adversary to select a wider range of disturbance combinations, it effectively enlarges the overall uncertainty faced by the agents, which leads to more conservative policies, as agents must hedge against more extreme worst-case scenarios. Our hierarchical uncertainty sets restrict the adversarial choices to a policy-dependent region, resulting in a smaller effective uncertainty set and therefore less conservative policies. This property is formally established in **Lemma 3.3** and **Theorem 3.4**.
>
> ---
>
> ### Q2. Eq 2 and Eq 3 are the same
>
> **A2**
>
> We thank the reviewer for pointing this out. Eq. (2) and Eq. (3) are intended to describe closely related quantities: Eq. (2) defines the expected return $J_i(s,\pi)$ under policy $\pi$, while Eq. (3) introduces the value function notation $V_i(s)$ for the same quantity starting from state $s$. We unify the notation and present a single definition to improve clarity (marked blue).
>
> ---
>
> ### Q3. Typo in Eq.6
>
> **A3**
>
> We thank the reviewer for catching this typo. In Eq. (6), the transition probability should be $\bar{P}(s' \mid s,a)$ instead of $\tilde{P}(s' \mid s,a)$. We have corrected this in the revision.
>
> ---
>
> ### Q4. Action sampled in Eq.9
>
> **A4**
>
> In Eq. (9), the notation $\bar a \sim \bar \pi$ is intended to indicate that the conditioned actions are induced by the worst-case perturbed policy $\bar\pi$. The uncertainty set in Eq. (9) is defined at the **state–action level**. In particular, once an action $\bar a$ is generated from the perturbed policy $\bar\pi$ (i.e., $\bar a \sim \bar\pi$), the environmental uncertainty set constrains admissible transition models $\bar P(s' \mid s,\bar a)$ for this specific state–action pair. Therefore, the set is defined with respect to the sampled action $\bar a$, rather than the entire action distribution induced by $\bar\pi$.
>
> ---
>
> ### Q5. Theorem 3.6 Proof / lack of dependence on $\delta$
>
> **A5**
>
> We thank the reviewer for pointing this out. In the proof of Theorem 3.6, the transition model should explicitly depend on the policy perturbation through the hierarchical uncertainty sets. The current notation $\tilde P(s'|s,a)$ omits this dependency for brevity and should be written as $\tilde P^H(s'|s,a,\delta)$ (or equivalently $\tilde P_s(\bar{\pi})$ with $\bar{\pi}=\pi+\delta$). We have corrected this typo and clarified the dependency in the revised version.
>
> Here $\pi(a|s)$ denotes the joint policy induced by decentralized policies, which factorizes as
>
> $\pi(a|s)=\prod_{j=1}^N \pi_j(a_j|s)$.
>
> In the proof, we rewrite this expression by separating the policy of agent $i$:
>
> $\pi(a|s)=\pi_i(a_i|s)\prod_{j\neq i}\pi_j(a_j|s)$.
>
> The summation over joint actions appears when evaluating expectations over $a$. We will clarify this step in the revision to make the derivation clearer.
>
> ---

---

> ### Author Response · Authors · 2026-03-16
> **Reply [2/2]**
>
> ### Q6. Equation 23: please clarify the implication of this approximation. It seems to me that you are implicitly assuming gaussian environmental noise.
>
> **A6**
>
> Eq. (23) indeed uses the closed-form differential entropy of a Gaussian distribution. In our implementation, we assume the environmental transition noise to be Gaussian, which allows us to compute the entropy term in a tractable form. This assumption is used only for estimating the environmental uncertainty in practice and does not affect the theoretical formulation of the hierarchical uncertainty sets. We have clarified this modeling assumption in the revision.
>
> ---
>
> ### Q7. Equations 25–27: why are you using the notation P instead of the joint policy already introduced?
>
> **A7**
>
> We thank the reviewer for pointing this out. In Eq. (25)–(27), $P(a|s)$ denotes the joint action distribution induced by the decentralized policies. For consistency with the notation used earlier in the paper, this distribution should be written as the joint policy $\pi(a|s)$. We have revised the notation accordingly to avoid confusion.
>
> ---
>
> ### Q8. The work mentions theoretical guarantees on the convergence of the actor-critic algorithms. Where are the convergence guarantees explicited in the work?
>
> **A8**
>
> In our original manuscript, **Theorem 3.2** establishes that the robust Bellman operator $T$ is a $\gamma$-contraction under the sup-norm. This result implies the existence and uniqueness of the robust value function and, in tabular settings, guarantees that the value iteration
>
> $V_{k+1}=T[V_k]$
>
> converges to the fixed point from any initialization.
>
> To further clarify the theoretical guarantees, we have added **Theorem 3.7** in the revised manuscript (highlighted in blue with proof in Appendix B.6), which shows that the robust Q-value also converges under the same tabular setting. This complements the contraction result for the value function and provides a theoretical justification for the critic update used in our algorithm.
>
> ---
>
> ### Q9. Please improve the writing in Section 3.3.3 when introducing the critic and actor updates. Currently, the introduction of both formulas are not well integrated, breaking the flow of the writing.
>
> **A9**
>
> We thank the reviewer for this helpful suggestion. In the revised manuscript, we have added additional transitional explanations (highlighted in blue) to better connect the critic and actor update formulas and improve the overall flow of the section.

---

> > ### Comment · Reviewer_hmJZ · 2026-03-17
> > **Reply [1/2]**
> >
> > I thank the authors for their answers and the multiple revisions that were provided.
> >
> > Please find below additional comments.
> >
> > **A1**
> >
> > Thank you for clarify this, it was my understanding of “max-of-both”. I understand this could lead to overly conservative policies, this is always the issue when considering guarantees over worst-case disturbances.
> >
> > A hierarchical dependency could indeed lead to less conservative policies, considering that the set of disturbances is a subset of the general set (Lemma 3.3 in the paper, see also my comments below).
> >
> > Why is this specific dependency natural? How does it remove the “bad” cases from the general formulation? As I understand, the definition of the perturbed environment depends on the joint action that is actually taken by the players. I do not see how the value function could be impacted by actions not played, and thus how Theorem 3.4 is not an equality, intuitively. If policies are stochastic, it is still unclear to me how the environmental perturbations are defined, see my comments below.
> >
> > **A5**
> >
> > What expectation over $a$ are you referring to? Also see my comments below.

---

> > > ### Comment · Reviewer_hmJZ · 2026-03-17
> > > **Reply [2/2]**
> > >
> > > Please also find here additional comments about the revised version of the paper.
> > >
> > > - “This hierarchical design preserves the causal dependency between interaction and environmental uncertainty by conditioning environmental perturbations on the worst-case perturbed policy, rather than allowing independent maximization over both sources.”
> > > Why do we have a causal dependency for the dependencies? I understand the action is chosen before the transition occurs but why aren’t the noises independent in general?
> > > - Equation 4: how is $\delta_j$ defined? instead of minimizing with $\delta_i$, shouldn’t you minimize with all but $\delta_i$? How is $a_j$ defined? Shouldn’t you sum over all joint actions?
> > > - Theorem 3.2. I think what you want to say is that the values are fixed point of the Bellman operator. Equation 5 is the definition of the Bellman operator, then the statement should be that there exists a unique fixed point.
> > > - Equation 6 is not defined correctly. The interaction uncertainty depends on $a$ which is only defined inside the expectation of the environmental uncertainty.
> > > - Equation 7 is the set of perturbations, not perturbed policies. “Specifically, given the nominal policy $\pi_i$, the perturbed policy $\overline{\pi_i} = \pi_i + \delta_i$ belongs to”. How do you define the entropy, for example $H(a | s)$? I understand the entropy of a distribution over actions, but what do you mean by $H(a | s)$? Is it the entropy of a random variable $a$ (needs to be defined) following some distribution?
> > > - Equation 8 is confusing to me. Please define $a$ and $\overline{a}$ inside the set, or with a quantifier such as $\forall a$. If the set cannot be defined because $\overline{a}$ is sampled a priori, then I believe a more natural and valid notation would be that the uncertainty set depends on $\overline{a}$ directly instead of $\overline{\pi}$.
> > > - Equation 9 has the same problem. I understand that $U_H$ implicitly depends on the policy. $s’,s,a$ are not defined. I would understand if the environmental uncertainty set is defined as a set of transition kernels.
> > > - Appendix B.2. $U_G$ is defined with $\tilde{P}_s(\pi)$, not $\tilde{P}_s$. Please explain how you can come to the conclusion with this definition of $U_G$. Additionally, $\tilde{P}_s(\delta)$ is never defined, especially since $\delta$ cannot be a probability distribution (since $\pi + \delta$ must be), the entropy with respect to $\delta$ would have to be clarified.
> > > - Appendix B.3. Please clarify “because $\tilde{P}_s(\delta)$ is in $\tilde{P}_s$”, as mentioned in the previous point. Also, how is that related to the two cited works? In general the statement makes sense if Lemma 3.3 holds, since the right hand side is minimizing over a superset.
> > > - “Lemma 3.3 and Theorem 3.4 show that the hierarchical sets exclude unrealistic joint disturbances”. It would be clearer to show what are the elements in $U_G$ that are not in $U_H$ and argue why these are unrealistic.
> > > - Equation 13. Similar to one of the points above, $a_j$ is not defined in the optimization problem. Also, it seems that $\eta^*_j$ should be used instead of $\eta_j$
> > > - Appendix B.4. Similarly, $delta_j$’s and $a_j$’s are not defined in the set. Shouldn’t it be $\tilde{P}_s(\pi + \delta)$? You mentioned in a previous reply that tilde $P^H(s’|s, a, \delta)$ could equivalently be written as $\tilde{P}_s(\overline{\pi})$, but I thought $\tilde{P}_s(\overline{\pi})$ is the set containing all the possible perturbed kernels
> > > - Appendix B.4. The definition of $\pi$ is still confusing to me. First, the individual policies are defined as $\pi_i$ for $i$ the index of a player. This definition is overloaded by $\pi_1$ and $\pi_2$, further used to define $\pi’$. Then, I believe you wanted to write $\pi(a | s) = \prod_i^N \pi_i(a_i|s)$. Then, the sum over all actions is confusing, because $a$ is defined twice. Did you want to use a different terms like $a’$? Why do we have a sum? Then, what is the definition of $\pi_j(a_j | s; \pi_1)$? How do you condition on a policy?
> > > - Equation 23. Please change your notation for the entropy. $s’$ is a state and not a random variable. I would expect $H(S’ | s, a)$ or $H(\cdot | s, a)$ to denote the entropy. Similar for equation 25. Also please define the entropy terms earlier in the paper, before using them.
> > > - Equation 26. I think you should use $\pi_i(a_i | o_i)$ instead of $\pi(a_i | o_i)$.

---

> > > > ### Author Response · Authors · 2026-03-19
> > > > **Continued Reply [2]**
> > > >
> > > > **Q2.3. Theorem 3.2**
> > > >
> > > > We thank the reviewer for the suggestion.
> > > > In the revised manuscript, we explicitly characterize the robust value function as the fixed point of the Bellman operator:
> > > > $$
> > > > T[\bar V_i^\*] = \bar V_i^\*,
> > > > $$
> > > > and clarify that $\bar{V}_i^*$ is the unique fixed point of $T$.
> > > >
> > > > ---
> > > >
> > > > **Q2.3 Equation 6**
> > > >
> > > > The interaction uncertainty is defined based on the entropy term $H(a \mid s)$, which is already an expectation over the policy distribution. Specifically,
> > > > $$
> > > > H(a \mid s) = - \sum_{a \in \mathcal{A}} \pi(a \mid s) \log \pi(a \mid s),
> > > > $$
> > > > and thus the dependence on $a$ is implicitly captured through the policy distribution.
> > > >
> > > > We have clarified this point in the revised manuscript by explicitly providing the definition of $H(a \mid s)$ to avoid ambiguity.
> > > >
> > > > ---
> > > >
> > > > **Q2.4 Equation 7 (Entropy $H(a \mid s)$)**
> > > >
> > > > We thank the reviewer for this important clarification.
> > > >
> > > > In our formulation, $a$ is treated as a random variable drawn from the policy distribution, i.e., $a \sim \pi(\cdot \mid s)$. Therefore, the entropy term $H(a \mid s)$ refers to the entropy of the action distribution induced by the policy at state $s$, rather than a fixed action.
> > > >
> > > > We have clarified this definition in the revised manuscript.
> > > >
> > > > ---
> > > >
> > > > **Q2.5 Eq. 8 and Eq. 9**
> > > >
> > > > To improve clarity, we revise Eq.~(8) to explicitly anchor the constraint at each nominal action $a$, while taking expectation over the perturbed action distribution $\bar{a} \sim \bar{\pi}$:
> > > >
> > > > $$\tilde{\mathcal{P}}_s(\bar \pi) =\Big\\{\bar{P}^H(\cdot \mid s, a, \delta)\Big|\big|H(S' \mid s, a)- H\_{\bar P^H}(S' \mid s, \bar a ) \big| \le \\rho\_\{\\text{env}}, \forall a \sim \pi, \forall \bar{a} \sim \bar{\pi} \Big\\}.$$
> > > >
> > > > Accordingly, the hierarchical uncertainty set is updated as:
> > > > $$\mathcal{U}_H= \big\\{ (\bar P^H, \delta)\big| \delta \in \Delta\_{\pi}, \bar P^H(\cdot \mid s, a, \delta) \in \tilde{\mathcal{P}}_s(\bar \pi),\forall s \in \mathcal{S},\;\forall a \sim \pi \big\\}.$$
> > > >
> > > > We have updated the manuscript accordingly to clarify the roles of $a$ and $\bar{a}$.
> > > >
> > > > ---
> > > >
> > > > **Q2.6 Appendix B.2**
> > > >
> > > > The transition uncertainty set should depend on the (possibly perturbed) policy, i.e., $\tilde{P}_s(\pi)$ or $\tilde{P}_s(\bar{\pi})$, rather than $\tilde{P}_s(\delta)$. We have corrected this notation in the revised manuscript.
> > > >
> > > > Therefore, entropy is always defined with respect to the induced policy distribution (e.g., $H(a \mid s)$ under $\pi$ or $\bar{\pi}$), and not directly with respect to $\delta$.
> > > >
> > > > Appendix B.2 has been revised accordingly.
> > > >
> > > > ---
> > > >
> > > > **Q2.7 Appendix B.3**
> > > >
> > > > The notation $\tilde{P}_s(\delta)$ was a typo. As clarified in Appendix B.2, the transition uncertainty set should depend on the (possibly perturbed) policy, i.e., $\tilde{P}_s(\pi)$ or $\tilde{P}_s(\bar{\pi})$. We have corrected this notation consistently throughout the manuscript.
> > > >
> > > > This result is consistent with standard robust stochastic game formulations (e.g., Karde\c{s} et al.), where the value function is defined via a min--max Bellman operator over admissible transition kernels. Moreover, following the comparison principle (e.g., Xu et al.), restricting the uncertainty set leads to a less conservative (i.e., higher) worst-case value, which aligns with our theoretical result once $U_H \subseteq U_G$ holds.
> > > >
> > > > ---
> > > >
> > > > **Q2.8 providing examples for Lemma 3.3 and Theorem 3.4**
> > > >
> > > > We have provided an explicit illustrative example in our response to **Q2.1**, based on an autonomous driving scenario, to demonstrate which elements in $U_G$ are excluded by $U_H$. Please refer to A1 for details.
> > > >
> > > > ---
> > > >
> > > > **Q2.9 Equation 13**
> > > >
> > > > We thank the reviewer for pointing this out.
> > > >
> > > > Eq. (13) is written from a per-agent best-response perspective. The optimization is taken with respect to $(\pi_i, \eta_i)$, while other agents follow optimal strategies $(\pi_j^\*, \eta_j^\*)$ for $j \neq i$.
> > > >
> > > > Accordingly, $a_j$ denotes the action of agent $j$ sampled from its optimal perturbed policy, i.e., $a_j \sim \bar{\pi}_j^*$.
> > > >
> > > > We have revised Eq. (13) to clarify this point.
> > > >
> > > > ---
> > > >
> > > > **Q2.10 Appendix B.4**
> > > >
> > > > We thank the reviewer for the helpful comment.
> > > >
> > > > Both $\delta_j$ and $a_j$ correspond to optimal solutions of other agents, and should be written as $\delta_j^\*$ and $a_j \sim \bar \pi_j^\*$. We have corrected this in the revised manuscript.
> > > >
> > > > The notation $\tilde{P}_s(\delta)$ was a typo. The transition uncertainty set should depend on the policy, i.e., $\tilde{\mathcal{P}}_s(\pi)$ or $\tilde{\mathcal{P}}_s(\bar \pi)$. We have corrected this throughout Appendix B.4.

---

> > > > ### Author Response · Authors · 2026-03-19
> > > > **Continued Reply [3]**
> > > >
> > > > **Q2.11 Appendix B.4: summation / $\pi$ overload / conditioning**
> > > >
> > > > We thank the reviewer for pointing out these notational issues.
> > > >
> > > > - **Overloaded notation of $\pi$:**
> > > >   We distinguish between individual policies $\pi_i$ and policy profiles $\pi^{(k)} = (\pi^{(k)}_1, \dots, \pi^{(k)}_N)$.
> > > >
> > > > - **Joint policy definition:**
> > > > $
> > > >   \pi(a \mid s) = \prod_{i=1}^N \pi_i(a_i \mid s),
> > > > $
> > > >   and the previous summation was a typo and has been removed.
> > > >
> > > > - **Conditioning on policies:**
> > > >   We remove incorrect conditioning expressions. All typos have been corrected in the revision.
> > > >
> > > > ---
> > > >
> > > > **Q2.12  Equation 23**
> > > >
> > > > We agree that the entropy notation was not precise. Using $s'$ may cause confusion, as it denotes a state rather than a random variable. We have revised the notation to use $S'$:
> > > > $H(S' \mid s,a).$
> > > >
> > > > ---
> > > >
> > > > **Q2.13  Equation 26**
> > > >
> > > > **Answer.**
> > > > We thank the reviewer for pointing this out. This was a typographical error.
> > > >
> > > > We have corrected Eq.~(26) to use $\pi_i(a_i \mid o_i)$ for clarity and consistency with agent-specific policy notation.

---

> > > ### Author Response · Authors · 2026-03-19
> > > **Continued Reply [1]**
> > >
> > > We sincerely thank the reviewer for the thoughtful and highly professional comments.
> > > We truly appreciate the time and patience the reviewer devoted to carefully reading our paper.
> > > The insightful and constructive suggestions have greatly improved the clarity and rigor of our manuscript.
> > >
> > > We address the questions below. The corresponding revisions have been incorporated into the manuscript and are highlighted in red to facilitate reference.
> > >
> > > **Q2.1. Why is this specific dependency natural? How does it remove the “bad” cases from the general formulation? & Why do we have a causal dependency?**
> > >
> > > **(1) Why is the dependency natural?**
> > >
> > > We want to clarify that our hierarchical structure is developed based on the entropy decomposition:
> > > $$
> > > H(S' \mid s) = \mathbb{E}_{a \sim \pi(\cdot|s)}[H(S' \mid s,a)] + H(a \mid s).
> > > $$
> > >
> > > In the term $H(S' \mid s,a)$, the action $a$ is sampled from the policy, and the transition distribution depends on the executed action. Therefore, any perturbation to the policy will naturally propagate to the transition distribution.
> > >
> > > This induces a natural hierarchical structure: interaction disturbances (policy perturbations) are selected first, and environmental disturbances (transition perturbations) are then conditioned on the perturbed policy.
> > >
> > > ---
> > >
> > > **(2) How does this remove “bad” cases from the general formulation?**
> > >
> > > Consider an autonomous driving scenario where the ego vehicle interacts with a nearby vehicle.
> > >
> > > - The **interaction uncertainty** captures how aggressively the nearby vehicle behaves (e.g., frequent abrupt lane changes).
> > > - The **environmental uncertainty** corresponds to the road friction coefficient, which determines whether such maneuvers are physically feasible.
> > >
> > > Under the **hierarchical uncertainty set**, the environmental disturbance is selected *conditioned on* the interaction disturbance.
> > >
> > > - If interaction uncertainty is large (aggressive driving),
> > > - then the environmental uncertainty excludes extreme cases such as near-zero friction, since such conditions cannot support aggressive maneuvers.
> > >
> > > In contrast, under a **general (independent) uncertainty set**, the adversary selects disturbances independently. This can produce unrealistic joint cases, such as:
> > >
> > > a highly aggressive vehicle performing sharp lane changes on a road with nearly zero friction.
> > >
> > > ---
> > >
> > > **(3) Clarification on the role of the perturbed transition kernel**
> > >
> > > We clarify that the perturbed transition kernel is **not a fixed environment model**, but an adversarial decision variable chosen by the nature player, as in standard robust RL / MARL formulations.
> > >
> > > Specifically, our framework follows a min–max structure:
> > > - the **nature player** selects worst-case disturbances (policy + transition),
> > > - the **agents** optimize policies against these disturbances.
> > >
> > > Thus, $\bar{P}^H(s' \mid s,a,\delta)$ should not be interpreted as a fixed transition model. Specifically, under the hierarchical uncertainty set, once a policy disturbance $\delta$ is selected, the admissible transition perturbations $\bar{P}^H$ are restricted to those that are consistent with the induced action distribution. This effectively removes unrealistic joint disturbance pairs like the example we gave.
> > >
> > > ---
> > >
> > > **Q2.2: Eq.4 How is $\delta_j$ defined? Instead of minimizing with $\delta_i$, shouldn't you minimize with respect to all $\delta_j$? How is $a_j$ defined? Shouldn't the summation be taken over all joint actions?**
> > >
> > > We thank the reviewer for the insightful questions.
> > > $\delta_j$ corresponds to the interaction disturbance associated with agent $j \neq i$, while $\delta_i$ denotes the disturbance applied to the policy of agent $i$. In our formulation, the minimization is performed only over $\delta_i$, as we adopt a best-response perspective for agent $i$. The disturbances of other agents, $\{\delta_j\}$, are treated as part of the optimal strategies and are therefore denoted as $\delta_j^*$.
> > >
> > > Similarly, $a_j$ represents the action of agent $j \neq i$, which is sampled from its optimal policy $\bar \pi_j^\*$. As a result, the joint contribution of other agents is captured through the product term
> > > $$
> > > \prod_{j \neq i} \bar \pi_j^\*(a_j|s, \delta_j^\*),
> > > $$
> > > where $\bar \pi_j^\* = \pi_j^\* + \delta_j^\*$, and does not require explicit optimization.
> > >
> > > Regarding the summation, we only sum over $a_i \in \mathcal{A}_i$ because the value function $\bar{V}_i$ is defined under the assumption that other agents follow their optimal strategies.
> > >
> > > We have revised Eq.~(4) accordingly to clarify these points and highlight the changes in red.

---

### Review · Reviewer_Vvmm · 2026-03-08

**Summary Of Contributions:**

This paper proposes the concept of Aleatoric Robust Equilibrium (ARE) which first structures the disturbances in an multi-agent system as 1) environment uncertainty from stochastic dynamics and 2) interaction uncertainty from behavior of other agents. This paper proposes that selecting the maximum disturbances from both environmental and agentic disturbances leads to overly conservative policies for a situation that rarely occurs. The hierarchical entropy-based structure places a heavier emphasis on the interaction uncertainty and less on the environment uncertainty to mitigate this effect. The authors validate their concept on multi-agent particle environment and MuJoCo environments.

Strengths
- Analyzing disturbances in MARL into parts (aleatoric/environment vs agent's behaviors) is an interesting approach

Weakness
- Rationale for analyzing disturbances in MARL into parts could be better structured + justified. My suggestions and questions are listed below.
- The authors places heavier weight on behavioral disturbances than the environmental disturbances. While this may be true in some cases, this is always not the case.

**Additional Comments:**

N/A

**Audience:**

Yes

**Audience Explanation:**

Multi-agent systems are highly complex environment with variables and disturbances introduced from both the environment itself and other agents in the system. Individuals interested in multi-agent systems, robust systems would be interested in research topics related to the ones shown in this paper

**Claims And Evidence:**

No

**Claims Explanation:**

I am not convinced by the claim in which disturbances in a multi-agent system can be divided as linear combination of disturbances from environment and behavior of other agents in the system. The disturbances in the system is conditioned on both behavior and environment.

For example, in the case of the F1 Monaco Grandprix, the tracks are narrow and slow and overtaking rarely happens. Therefore, disturbances caused by other agents in the system, in conjunction with the environment, in this case has less effect on the ego agent's performance than other environment setups. In this case, it wouldn't be accurate to model the disturbances in a multiagent system as simply sum of disturbances from environment and disturbances from other agents in the system, but as disturbance conditioned on both the environment and other agents.

Second, while the paper claims that the most existing works only consider one type of disturbances but not both, several existing works consider both environment and other agent's behavior into MARL setup. MAESTRO (Samevylan et al, 2023) and GEMS (Song et al 2024) both are good examples of this case, with the later work being applied to both competitive and collaborative setup as well.

**Requested Changes:**

I request the authors to address the points made in the 'Are the claims made in the submission supported by accurate, convincing and clear evidence?'. 1) Justification on formulating disturbances in a multi-agent system as a combination, not a conditional function, of disturbances form the environment and other policies and 2) additional efforts on related works and explain how this paper is novel compared to existing works.

---

> ### Author Response · Authors · 2026-03-16
> **Reply [1/1]**
>
> Many thanks for your careful evaluation and highly constructive comments.
> The questions raised have helped us further strengthen the clarity and completeness of the work.
> We provide detailed, point-by-point responses below.
>
> ---
>
> ### Q1
> I am not convinced by the claim in which disturbances in a multi-agent system can be divided as linear combination of disturbances from environment and behavior of other agents in the system. The disturbances in the system is conditioned on both behavior and environment.
>
> **A1**
>
> We thank the reviewer for raising this point. We would like to clarify that our formulation does **not assume that disturbances are generated as a linear combination** of environmental and behavioral perturbations.
>
> The decomposition in Eq. (7) follows directly from the **chain rule of entropy**
>
> $$
> H(s'|s) = \mathbb{E}_{a \sim \pi}[H(s'|s,a)] + H(a|s),
> $$
>
> which is an **information-theoretic identity** rather than an additive disturbance model. The purpose of this decomposition is to characterize two sources of aleatoric uncertainty:
>
> - uncertainty arising from stochastic environment dynamics ($H(s'|s,a)$), and
> - uncertainty arising from stochastic agent behaviors ($H(a|s)$).
>
> We agree with the reviewer that in real multi-agent systems these two sources are often coupled. Our framework explicitly accounts for this dependency through the **hierarchical uncertainty sets**. In particular, the environmental uncertainty set $\bar{P}^H(\delta)$ is conditioned on the perturbed policy signal $\delta$, meaning that environmental perturbations depend on the interaction disturbances rather than being optimized independently.
>
> Therefore, the goal of our work is to analyze the different sources of aleatoric uncertainty in multi-agent systems and provide a framework to quantify their respective contributions. While these uncertainties may be coupled in practice, our framework models this dependency through the proposed hierarchical uncertainty sets, allowing us to study and quantify different sources of uncertainty while still capturing their inherent coupling.
>
> ---
>
> ### Q2
> Additional efforts on related works and explain how this paper is novel compared to existing works.
>
> **A2**
>
> We thank the reviewer for pointing out the related works **MAESTRO (Samvelyan et al., 2023)** and **GEMS (Song et al., 2024)**.
>
> We would like to clarify that our statement specifically refers to the literature on **robust MARL and uncertainty-aware MARL**. In this line of work, most existing approaches typically consider only a single type of disturbance, such as transition uncertainty or adversarial policy perturbations.
>
> MAESTRO and GEMS address a different research problem. These works focus on curriculum generation and population-based training, where the learning process adapts the distribution of environments and co-player policies during training. For example, MAESTRO explicitly constructs a *joint curriculum over environments and co-players* to improve generalization. However, these approaches do not model uncertainty sets or analyze the structure of uncertainty in the underlying Markov game.
>
> In contrast, our work focuses on uncertainty modeling rather than curriculum generation. We explicitly analyze the sources of aleatoric uncertainty in MARL and formulate hierarchical uncertainty sets to characterize environmental and interaction uncertainties within a robust Markov game framework.
>
> To avoid confusion, we revised the manuscript in the *Related Work* section to clarify that our statement refers to the **robust MARL literature** rather than the broader MARL literature.

---

> > ### Comment · Reviewer_Vvmm · 2026-04-07
> >
> > I thank the authors for the clarification and addressing the question. I have no further concerns.

---

### Review · Reviewer_H7mZ · 2026-03-08

**Summary Of Contributions:**

The authors present an approach to construct less conservative, principled solutions for MARL algorithms under uncertainty, resulting in actor-critic methods that are better performing than existing baselines for learning policies in robust Markov games. To do this, they argue that uncertainty in state transitions (quantified as the entropy of the next state conditioned on the current state) can be hierarchically decomposed as the expected entropy over the actions sampled from the policy plus the policy entropy. This allows the authors to derive restricted uncertainty sets that lead to smaller joint adversarial nature policy sets, and therefore better returns for the learned agent policies. The authors present theoretical results on the ARE existence, and a thorough experimental evaluation on MARL problems with noise/uncertainty, showing better performance than existing baselines, with adequate ablation studies.

## Strengths
- The uncertainty decomposition is comprehensive and intuitive, and the case against the "max-of-both" formulation is convincing.
- The experimental results are very thorough, and the uncertainty decomposition analysis provides interesting insights as to how injected noise affects the different uncertainty levels.
- In general the paper is coherent and the work is well justified.

## Weaknesses
- Some statements and assumptions are not clear (see below for questions).
- The computational overhead of the method when compared to existing baselines is not really discussed.
- The authors state that 'convergence guarantees' are provided, but convergence of the method is not discussed (unless I missed some result).

**Audience:**

Yes

**Audience Explanation:**

The problem of uncertainty in MARL is very relevant and I believe that the paper provides interesting insights, and the evaluation is thorough enough to back the claims.

**Broader Impact Concerns:**

No concerns.

**Claims And Evidence:**

Yes

**Claims Explanation:**

Relatively, subject to clarifications and questions below.

**Requested Changes:**

I include here some questions, comments and clarity issues.
- Are $\mathcal{U}_H$ and $\mathcal{U}_G$ supposed to be defined for the same maximum common uncertainty level? I understand the point is then that a Markov game under that common uncertainty level can be solved using U_H (and the corresponding decomposition) and one finds less conservative policies. This assumes that nature 'cannot' just pick from the product of uncertainty sets, but it's still not clear to me whether this is an assumption or a causal fact.  The authors justify this by saying action-selection randomness affects future state randomness, but in many systems the opposite happens: changes in environment dynamics first alter what other agents do, so interaction uncertainty may be a cause of of environmental uncertainty (agents may randomise actions because precisely the transitions are uncertain). This is not really clear to me yet.
- $\mathcal{U}_G$ is introduced without detail on how it relates to $\mathcal{U}_H$. Could the authors clarify how $\mathcal{U}_G$ is defined, especially with respect to the maximum $\rho_{env}$ and $\rho_{int}$ that define $\mathcal{U}_H$?
- The authors state in the claims that the resulting algorithms have convergence guarantees. The results in Theorem 3.6 show the existence of the ARE. Is there a convergence guarantee in tabular settings that extends to the author's results?
- The selection or tuning of $\rho_{env}$ and $\rho_{int}$ is not really discussed.
- The authors cite many recent works that are not really included in the experimental baselines (eg Bukharin et al). Is there a reason for this?
Some notation is confusing:
- $P$ and $\mathcal{P}$ are both used for transition functions (probability kernels), but then $\mathcal{P}$ is used in eq 9 to define a set.
- Equations 8 and 9 include $a,s$ in the definition of the sets. Are these defined for all $a,s$? Or is there one set defined per pair?
- Eq 12 infimum is taken for $\mathcal{P},\delta$ in the sets $\mathcal{U}_G$, $\mathcal{U}_H$?
- Definition 3.5 uses $\rho^*$ for the environment optimal policy but $\rho$ is used earlier for the maximum uncertainty levels.

---

> ### Author Response · Authors · 2026-03-16
> **Reply [1/2]**
>
> We sincerely thank the reviewer for taking the time to review our paper. We thank the reviewer for giving detailed and constructive comments. We have revised the paper according to your suggestions, and the responses to your comments are listed below.
>
> ---
>
> ### Q1.
> Are $\mathcal{U}_H$ and $\mathcal{U}_G$ defined for the same maximum uncertainty level?
> If so, does the proposed approach implicitly assume that nature cannot select disturbances independently from the product of uncertainty sets?
> It is not clear whether this restriction reflects a modeling assumption or a causal property of the system.
>
> **A1.1**
> Yes, both $U_H$ and $U_G$ are defined under the same uncertainty budgets $\rho_{\text{env}}$ and $\rho_{\text{int}}$.
> The difference is not the magnitude of uncertainty, but how the adversarial disturbances are selected.
>
> $\mathcal{U}_G$ allows nature to independently select the worst-case environmental and interaction disturbances, while $\mathcal{U}_H$ introduces a hierarchical selection where environmental disturbances are conditioned on the perturbed policy (interaction disturbances).
>
> **A1.2**
> We want to clarify that our hierarchical structure is an **assumption**, and it is developed based on the equation:
>
> $$
> H(s' \mid s) = \mathbb{E}_{a \sim \pi(\cdot|s)}[H(s' \mid s,a)] + H(a \mid s).
> $$
>
> This decomposition shows that the uncertainty of the next state arises from two sources:
>
> (i) the stochasticity of the policy through the action distribution $H(a|s)$, and
> (ii) the stochasticity of the environment conditioned on the selected action $H(s'|s,a)$.
>
> Because in the term $H(s'|s,a)$ actions are sampled from the policy and the transition distribution depends on the actions already performed, policy perturbations naturally propagate to the transition distribution.
>
> This explains the hierarchical structure used in our uncertainty sets, where interaction disturbances are selected first and environmental disturbances are conditioned on the perturbed policy.
>
> We agree that in many real-world systems, changes in the environment can influence agent behavior. Therefore, the causal structure where environmental uncertainty affects action selection is indeed common. However, the corresponding hierarchical uncertainty set would take the form
>
> $$\mathcal{U}_H^{rev} =\\{(\bar{P}, \delta)\mid \bar{P}  \in \tilde{P}_s, \delta  \in \Delta _{\pi}(\bar{P})\\} $$
>
> In this case, the associated decomposition would no longer be based on $H(s'|s)$ as in our paper, but rather on the reverse-order chain rule of the joint uncertainty:
>
>
> $$
> H(a,s' \mid s) = H(s' \mid s) + H(a \mid s,s').
> $$
>
> This corresponds to a different uncertainty structure, where policy uncertainty is conditioned on the realized transition uncertainty.
>
> ---
>
> ### Q2.
> $\mathcal{U}_G$ is introduced without detail on how it relates to $\mathcal{U}_H$.
>
> Could the authors clarify how $\mathcal{U}_G$ is defined,
>
> especially with respect to the maximum $\rho_{\text{env}}$
> and $\rho_{\text{int}}$
> that define $\mathcal{U}_H$?
>
> **A2**
>
> We thank the reviewer for pointing this out. The general uncertainty set $\mathcal{U}_G$ is defined using the same uncertainty
>
> budgets $\rho_{\text{env}}$ and $\rho_{\text{int}}$ as those used in $\mathcal{U}_H$.
>
> Specifically, we define the interaction and environmental uncertainty sets as
>
> $$ \Delta_{\pi}=\\{\delta \mid H(a \mid s) - H_{\bar{\pi}}(a \mid s) \le \rho_{\text{int}}\\}$$
>
>
> $$\tilde{P}_s =  \bar{P} (s' \mid s, \bar{a}) \mid H(s' \mid s,a) - H\_\bar{P}(s' \mid s, \bar{a}) \le \rho\_{\mathrm{env}}$$
>
> The general uncertainty set is then defined as
>
> $$
> \mathcal{U}\_G = \\{(\bar{P},\delta) \mid \delta \in \Delta_{\pi}, \bar{P} \in \tilde{P}_s \\}.
> $$
>
> Under $\mathcal{U}_G$, interaction perturbations $\delta$ and environmental perturbations $\bar{P}$ can be selected independently by the adversary.
>
> This definition is illustrated in Eq. (10) of the revised manuscript, which is highlighted in blue for clarity.
>
> ---
>
> ### Q3.
> The authors state in the claims that the resulting algorithms have convergence guarantees. The results in Theorem 3.6 show the existence of the ARE. Is there a convergence guarantee in tabular settings that extends to the author's results?
>
> **A3**
>
> In our original manuscript, Theorem 3.2 establishes that the robust Bellman operator $T$ is a $\gamma$-contraction under the sup-norm. This result implies the existence and uniqueness of the robust value function and, in tabular settings, guarantees that the value iteration
>
> $$
> V_{k+1} = T[V_k]
> $$
>
> converges to the fixed point from any initialization.
>
> To further clarify the theoretical guarantees, we have added Theorem 3.7 in the revised manuscript (highlighted in blue with proof in Appendix B.6), which shows that the robust Q-value also converges under the same tabular setting. This complements the contraction result for the value function and provides a theoretical justification for the critic update used in our algorithm.

---

> ### Author Response · Authors · 2026-03-16
> **Reply [2/2]**
>
> ### Q4.
> The selection or tuning of $\rho_{\text{env}}$ and $\rho_{\text{int}}$ is not really discussed.
>
> **A4**
>
> In our formulation, $\rho_{\text{env}}$ and $\rho_{\text{int}}$ represent the uncertainty budgets that control the size of the environmental and interaction uncertainty sets. Their selection follows the standard practice in uncertainty-set-based robust RL methods.
>
> In particular, several robust MARL baselines define uncertainty directly at the transition level. For example, methods such as R-MAD (one of our baselines) parameterize environmental uncertainty as perturbations of the transition probability
>
> $$
> \bar{P} = P + \delta
> $$
>
> where larger $\delta$ corresponds to a larger uncertainty set. In these works, experiments are typically conducted under different uncertainty regimes (e.g., low, medium, and high uncertainty).
>
> To ensure a fair comparison with such baselines, we calibrate our entropy-based uncertainty budgets so that the induced uncertainty levels correspond to the same regimes used in these prior methods. Specifically, $\rho_{\text{env}}$ is chosen to match the low/medium/high uncertainty levels used by the baselines. Since interaction uncertainty is defined in the same entropy units, we set $\rho_{\text{int}}$ at the same scale as $\rho_{\text{env}}$ in our experiments.
>
> We further clarify these details in Appendix C.3.
>
> ---
>
> ### Q5.
> The authors cite many recent works that are not really included in the experimental baselines (eg Bukharin et al). Is there a reason for this?
>
> **A5**
>
> Existing uncertainty-aware MARL methods can broadly be divided into two categories.
>
> The first category models uncertainty through explicit uncertainty sets and solves the resulting problem via max–min optimization. Representative examples include R-MAD and M3DDPG, as well as our proposed framework. In these approaches, robustness is defined with respect to adversarial perturbations within a specified uncertainty set.
>
> The second category incorporates uncertainty through regularization terms added to the objective (e.g., entropy or variance-based penalties). These methods do not explicitly define uncertainty sets but instead encourage robustness via soft regularization.
>
> Because the uncertainty representations in these two categories are fundamentally different, it is difficult to calibrate them in a comparable way. In particular, there is no clear correspondence between the size of an uncertainty set (e.g., the budget $\rho$ in our formulation) and the strength of a regularization coefficient in the objective. As a result, constructing a fair comparison where both approaches operate under equivalent uncertainty levels is non-trivial.
>
> ---
>
> ### Q6.
> $P$ and $\tilde{P}$ are both used for transition functions, but then $\mathcal{P}$ is used in Eq. (9) to define a set.
>
> **A6**
>
> We agree that the notation involving $P$ was dense in the previous manuscript.
>
> In the revised manuscript we clarified the notation as follows:
>
> - $P(s'|s,a)$ denotes the nominal transition dynamics
> - $\bar{P}(s'|s,a)$ denotes a perturbed transition kernel selected by the adversary
> - $\bar{P}^H(s'|s,a,\delta)$ denotes the hierarchical perturbed transition kernel conditioned on the interaction perturbation $\delta$
> - $\tilde{\mathcal{P}}_s(\bar{\pi})$ denotes the environmental uncertainty set containing admissible perturbed transition kernels
>
> To further improve readability, we also added a notation summary in Appendix A listing all symbols used in the paper.
>
> ---
>
> ### Q7.
> Equations 8 and 9 include $a$ and $s$ in the definition of the sets. Are these defined for all $(s,a)$ or is there one set per pair?
>
> **A7**
>
> In Eq. (8) and (9), the uncertainty constraints are defined in a state-wise (and state–action-wise) manner.
>
> Specifically, the interaction uncertainty in Eq. (8) is defined with respect to state $s$ since it is based on the conditional entropy $H(a|s)$. Thus, the constraint is applied state-wise for all $s$.
>
> Similarly, the environmental uncertainty in Eq. (9) is defined with respect to the state–action pair $(s,a)$ through the entropy $H(s'|s,a)$.
>
> ---
>
> ### Q8.
> In Eq. (12), is the infimum taken over $(\tilde P,\delta)$ within the uncertainty sets $U_G$ and $U_H$?
>
> **A8**
>
> We thank the reviewer for this clarification. In Eq. (12), the infimum is indeed taken over the perturbation pair $(\bar{P},\delta)$ contained in the corresponding uncertainty sets.
>
> More precisely,
>
> $$
> \inf_{(\bar{P}^H,\delta)\in U_H} V_{\bar{P}^H,\delta}(s)
> \ge
> \inf_{(\bar{P},\delta)\in U_G} V_{\bar{P},\delta}(s).
> $$
>
> We clarified this notation in the revised manuscript.
>
> ---
>
> ### Q9.
> Definition 3.5 uses $\rho^*$ for the environment optimal policy but $\rho$ is used earlier for the maximum uncertainty levels.
>
> **A9**
>
> In the revised manuscript we replaced the symbol $\rho^*$ with a different notation $\eta$ to clearly distinguish it from the uncertainty budget parameters and avoid potential ambiguity.

---

> > ### Comment · Reviewer_H7mZ · 2026-03-31
> >
> > I thank the authors for the detailed reply. I have no further concerns.

---

### Author Response · Authors · 2026-03-16
**General Response**

We sincerely thank all reviewers for their careful reading of our manuscript and for the insightful and constructive feedback. The comments were very helpful and have improved the clarity of our paper.

In response to the reviewers’ suggestions, we have revised the manuscript accordingly. The main changes are summarized below.

**1. Manuscript revisions.**
Following the reviewers' suggestions, we revised the manuscript and highlighted all modifications in blue to facilitate comparison.

**2. Notation clarification.**
Several notations were revised and clarified since there were some unclear points in the original description. In particular, we refined the transition notation as follows:

- $P(s'|s,a)$ denotes the nominal transition dynamics
- $\bar{P}(s'|s,a)$ denotes a perturbed transition kernel selected by the adversary
- $\bar{P}^H(s'|s,a,\delta)$ denotes the hierarchical perturbed transition kernel conditioned on the interaction perturbation $\delta$
- $\tilde{\mathcal{P}}_s(\bar{\pi})$ denotes the environmental uncertainty set containing admissible perturbed transition kernels

In addition, the parameter $\rho*$ in Eq. (13) has been replaced by $\eta^*$ to avoid confusion.
We also added a notation table in Appendix A to summarize the main symbols used in the paper.

**3. Additional theoretical results.**
To further support the critic learning convergence, we added a convergence proof for the robust state–action value function (Q-function) in Theorem 3.7 with proof in Appendix B.6.

Also, all the other detailed responses to each reviewer are provided below.

---

### Comment · Action_Editor_droh · 2026-04-22

Dear Authors,

Thank you for submitting your work to TMLR and for engaging extensively in the discussion.

There is some disagreement between the reviewers on the paper's evaluation, so I wanted to take some time to have a look at the work before taking a decision. I apologize for the resulting delay in the process.

I am not sure I have understood how the uncertainty sets are defined, especially by looking at eq. 8, 9. First, it looks odd to me to constrain the perturbation on the entropy of the distributions rather than some distance/divergence. Even if the $\rho_{int}$, $\rho_{env}$ are zero, there is several distributions with the same entropy. Let's say the policy $\bar \pi_i$ is described as $[1, 0, 0]$, then the perturbed policy can be any of $[0, 1, 0], [0, 0, 1]$? Can the Authors clarify whether I am interpreting the expressions correctly.

Moreover, I still do not understand where are the $s$ coming from. Are the uncertainty sets defined for any state? For states coming from some distribution?

I would like the authors to clarify those doubts before going ahead with the process.

Best regards,

AE

---

> ### Author Response · Authors · 2026-04-23
> **Reply [1/1]**
>
> We sincerely thank the Action Editor for the careful reading of our manuscript and for the thoughtful and constructive questions. We truly appreciate the time and effort devoted to understanding our work in depth, especially in pointing out aspects that could benefit from further clarification.
>
> ---
>
> **Q1.** Even if $\rho_{\text{int}}$ and $\rho_{\text{env}}$ are zero, there may exist multiple distributions with the same entropy. For example, if the policy $\pi_i$ is given by $[1,0,0]$, then the perturbed policy could be $[0,1,0]$ or $[0,0,1]$.
>
> **A1.** Thanks for the insightful question. We agree that if disturbances are allowed to be **arbitrarily large**, transitions such as $[1,0,0] \rightarrow [0,1,0]$ or $[0,0,1]$ could indeed occur under an entropy-based constraint alone. However, this is not the intended interpretation of our formulation.
>
> In our framework, the disturbances are not arbitrary, but represent *admissible perturbations* around a nominal policy and nominal transition model. This assumption is standard in robust MARL and robust control [1, 2], where uncertainty is modeled as perturbations around a nominal distribution. In practice, this nominal distribution is often taken as Gaussian (as also discussed in *Reviewer hmJZ, Q6*). As a result, the uncertainty set is further restricted by the assumption that perturbations arise from a nominal (e.g., Gaussian) reference model. This ensures that the admissible perturbations do not include arbitrary/radical distributions.
>
> In particular, transitions such as $[1,0,0] \rightarrow [0,1,0]$ do not correspond to admissible perturbations under our modeling assumption. Instead, typical perturbations introduce moderate stochasticity around the nominal policy. For example, a distribution such as $[0.6,0.4,0]$ may be perturbed to $[0.5,0.35,0.15]$, rather than a complete shift between actions.
>
> To improve clarity and rigor, we will revise the manuscript  to explicitly state that the uncertainty sets are defined with respect to nominal distributions, and that perturbations are admissible rather than arbitrary.
>
> ---
>
> **Q2.** Are the uncertainty sets defined for any state? For states coming from some distribution?
>
> **A2.** The uncertainty sets in our formulation are defined *pointwise* for each state $s \in \mathcal{S}$.
>
> To make this explicit, we have revised Eq. (7) and (8) as follows:
>
> $$ \Delta_{\pi_i} =\big\\{\delta_i \big\|H(a \mid s)- H_{\bar{\pi}_i}(\bar{a} \mid s) \le \rho\_{\text{int}}, \forall s \in \mathcal{S},\forall a \sim \pi, \forall \bar{a} \sim \bar{\pi} \big\\}.$$
>
> $$ \tilde{\mathcal{P}}_s(\bar{\pi}) = \big\\{ \bar{P}^H(\cdot \mid s,a,\delta)\big|  |H(S' \mid s,a)-H\_{\bar{P}^H}(S' | s,\bar a)| \le \rho\_{\text{env}},\forall s \in \mathcal{S},\forall a \sim \pi, \forall \bar{a} \sim \bar{\pi}\big\\}.$$
>
> A similar concern was also raised by Reviewer hmJZ (Q7),  please kindly refer that discussion for completeness.

---

> > ### Comment · Action_Editor_droh · 2026-04-24
> >
> > Dear Authors,
> >
> > Thank you for providing clarifications on my questions above.
> >
> > My current understanding is that there are two kinds of disturbances constraints at play here, one on the magnitude of the disturbance, the other on the entropy of the disturbed distribution.
> >
> > My concern is that the first type of constraint is essentially implicit (or at least, I couldn't find it stated anywhere). This is especially problematic in my view, because the whole paper is built on the definitions of the uncertainty sets. If those are not rigorously defined, I am worried that the rest of the paper cannot be evaluated thoroughly.
> >
> > For this reason, I am providing a recommendation of "reject with suggestion to resubmit". I really believe this work will be a nice contribution to TMLR, after a further pass to make the formulation rigorous.
> >
> > Some notes that may be useful for future versions:
> > - If the intended disturbance constraint is on the magnitude of the disturbance, why not formulating it as distance/divergence between the nominal and disturbed policy/transition? If a bound on the difference of the entropy is really needed, i think that can be obtained as a by-product in many cases, e.g., with a KL divergence;
> > - Note that in eq. 7 the constraint is on the signed difference. This means that the entropy of the perturbed policy can be arbitrarily large. Perhaps a typo?
> >
> > Best regards,
> >
> > AE

---

### Decision · Action_Editor_droh · 2026-04-24

**Recommendation:** Reject

**Additional Comments:**

I am providing a recommendation of "reject with suggestion to resubmit" sue to some concern over the rigour of the formulation. I really believe this work will be a nice contribution to TMLR, after a further pass to make the formulation clearer. Unfortunately, I believe the new version will require an additional round of review to be properly evaluated.

Some notes that may be useful for future versions:

If the intended disturbance constraint is on the magnitude of the disturbance, why not formulating it as distance/divergence between the nominal and disturbed policy/transition? If a bound on the difference of the entropy is really needed, i think that can be obtained as a by-product in many cases, e.g., with a KL divergence;
Note that in eq. 7 the constraint is on the signed difference. This means that the entropy of the perturbed policy can be arbitrarily large. Perhaps a typo?

**Audience:**

Yes

**Audience Explanation:**

The paper may draw the interest of the communities of multi-agent learning and robust learning.

**Claims And Evidence:**

No

**Claims Explanation:**

The claims in the paper are heavily based on a formulation of the uncertainty sets that may be underspecified. This makes it hard for reviewers to fully evaluate the claims.

See my comment in the discussion below:

> My current understanding is that there are two kinds of disturbances constraints at play here, one on the magnitude of the disturbance, the other on the entropy of the disturbed distribution. My concern is that the first type of constraint is essentially implicit (or at least, I couldn't find it stated anywhere). This is especially problematic in my view, because the whole paper is built on the definitions of the uncertainty sets. If those are not rigorously defined, I am worried that the rest of the paper cannot be evaluated thoroughly.

**Resubmission Of Major Revision:**

The authors may consider submitting a major revision at a later time.